# Noise-robust Graph Learning by Estimating and Leveraging Pairwise Interactions

**Xuefeng Du**[*]                                                                    *xfdu@cs.wisc.edu*
*University of Wisconsin-Madison*

**Tian Bian**[*]                                                                    *tbian@se.cuhk.edu.hk*
*The Chinese University of Hong Kong*

**Yu Rong**                                                                        *yu.rong@hotmail.com*
*Tencent AI Lab*

**Bo Han**                                                                        *bhanml@comp.hkbu.edu.hk*
*Hong Kong Baptist University*

**Tongliang Liu**                                                                    *tongliang.liu@sydney.edu.au*
*Mohamed bin Zayed University of Artificial Intelligence*
*The University of Sydney*

**Tingyang Xu**                                                                    *yu.rong@hotmail.com*
*Tencent AI Lab*

**Wenbing Huang**                                                                    *hwenbing@126.com*
*Renmin University of China*

**Yixuan Li**                                                                        *sharonli@cs.wisc.edu*
*University of Wisconsin-Madison*

**Junzhou Huang**                                                                    *jzhuang@uta.edu*
*University of Texas at Arlington*

**Reviewed on OpenReview:** *https://openreview.net/forum?id=r7imkFEAQb*

## Abstract

Teaching Graph Neural Networks (GNNs) to accurately classify nodes under severely noisy labels is an important problem in real-world graph learning applications, but is currently underexplored. Although *pairwise* training methods have demonstrated promise in supervised metric learning and unsupervised contrastive learning, they remain less studied on noisy graphs, where the structural pairwise interactions (PI) between nodes are abundant and thus might benefit label noise learning rather than the *pointwise* methods. This paper bridges the gap by proposing a pairwise framework for noisy node classification on graphs, which relies on the PI as a primary learning proxy in addition to the pointwise learning from the noisy node class labels. Our proposed framework `PI-GNN` contributes two novel components: (1) a confidence-aware PI estimation model that adaptively estimates the PI labels, which are defined as whether the two nodes share the same node labels, and (2) a decoupled training approach that leverages the estimated PI labels to regularize a node classification model for robust node classification. Extensive experiments on different datasets and GNN architectures demonstrate the effectiveness of `PI-GNN`, yielding a promising improvement over the state-of-the-art methods. Code is publicly available at `https://github.com/TianBian95/pi-gnn`.

---

[*]Xuefeng and Tian contributed equally to this work. Work is done while interning at Tencent.

## 1 Introduction

Graphs are ubiquitously used to represent data in different fields, including social networks, bioinformatics, recommendation systems, and computer network security. Accordingly, graph analysis tasks, such as node classification, have a significant impact in reality (Xu et al., 2019). The success of machine learning models, such as graph neural networks (GNNs) on node classification relies heavily on the collection of large datasets with human-annotated labels (Zhou et al., 2019). However, it is extremely expensive and time-consuming to label millions of nodes with high-quality annotations. Therefore, when dealing with large graphs, usually a subset of nodes is labeled, and a wide spectrum of semi-supervised learning techniques have emerged for improving node classification performance (Zhu et al., 2003; Zhou et al., 2003; Kipf & Welling, 2017).

Although achieving promising results, these techniques overlook the existence of noisy node labels. For instance, practitioners often leverage inexpensive alternatives for annotation, such as combining human and machine-generated label (Hu et al., 2020), which inevitably yields samples with noisy labels. Since neural networks (including GNNs) are able to memorize any given (random) labels (NT et al., 2019; Zhang et al., 2017), these noisy labels would easily prevent them from generalizing well. Therefore, training robust GNNs for *semi-supervised node classification against noisy labels* becomes increasingly *crucial but less studied* for safety-critical graph analysis, such as predicting the identity groups of users in social networks or the function of proteins to facilitate wet laboratory experiments, etc.

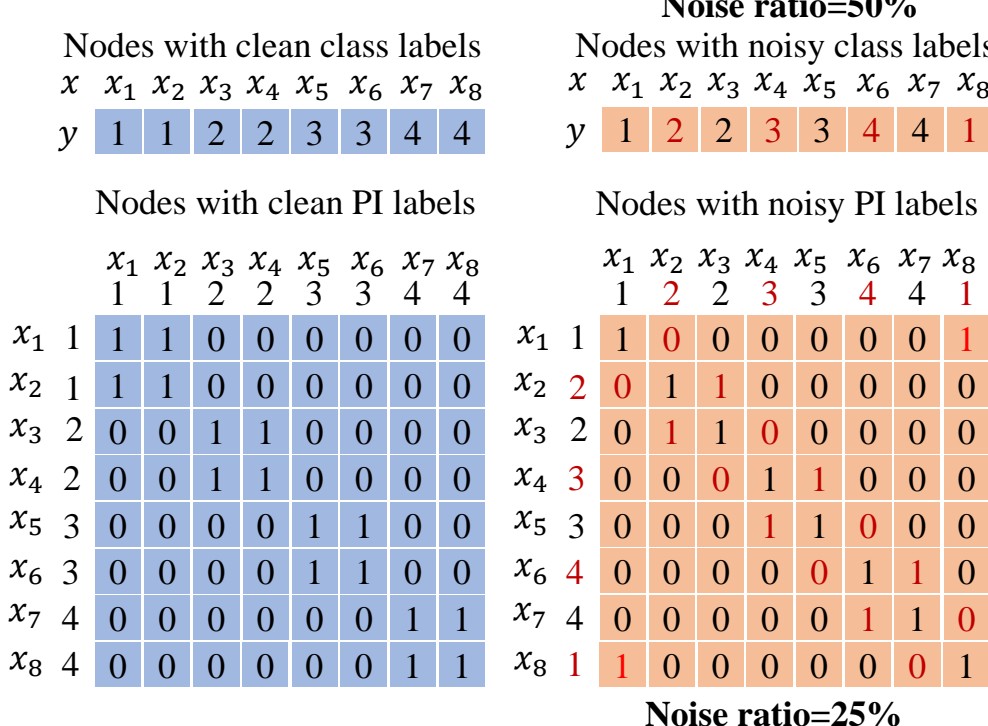

Figure 1: **Noise ratio comparison with noisy node labels**. The noise ratio of PI labels is much smaller than that of node labels. Number in red denotes noisy label.

In this paper, we pioneer a pairwise framework for noise-robust node classification on graphs, where relationships between data points are exploited. Currently, although the pairwise approaches are prevailing and made great progress in supervised metric learning and unsupervised contrastive learning (Qi et al., 2020; Boudiaf et al., 2020; Chen et al., 2020a; He et al., 2020), they remain largely unexplored in noise-robust graph learning. In particular, existing pointwise noise-robust learning algorithms (NT et al., 2019; Xia et al., 2021; Li et al., 2021a; Dong et al., 2020; Li et al., 2020a) are mainly designed for image inputs and strictly rely on the class label that shows the class that a node belongs to for learning. In contrast, the pairwise framework is

able to utilize the pairwise interactions (PI) between nodes, which indicate whether or not two nodes belong to the same class, as a learning proxy. As a result, it reduces the multi-class classification problem into a binary classification problem, which is easier to handle Patrini et al. (2017) and provides helpful learning signals apart from the noisy pointwise supervision. For example, Figure 1[1]. shows the transformation from the class labels to the PI labels. We can easily observe that the noise rate for the PI labels is much lower than that of the pointwise noisy class labels. Considering two nodes from the same class have the same noisy labels, their PI label still remains positive, which is helpful for the model to learn.

Although learning with PI intuitively demonstrates promise, it does not trivially transfer to label noise learning on graphs. For example, previous pairwise learning frameworks (Qi et al., 2020; Chen et al., 2020a) can easily calculate the PI labels either through class label comparison (same class label→positive PI label) or data augmentation (augmented views from the same image→positive PI label). However, PI labels can still contain unneglectable noise (*cf.* Figure 1) if we directly compare their noisy node class labels. As a result, the pairwise learning algorithm relying on such suboptimal PI labels can misbehave.

We propose a novel framework dubbed `PI-GNN`, tackling two highly dependent problems—PI estimation and learning—in one synergistic framework. Concretely, `PI-GNN` contributes two novel components: **(1)** We introduce an end-to-end confidence-aware PI label estimation branch that dynamically estimates PI labels with the help of graph structure (Section 3.1). In particular, we learn a graph neural network that is trained to predict node connectivity, where the connected nodes have a ground truth of 1 and vice versa. Compared to using node connectivity as the PI label directly, *i.e.*, connected nodes transform to a positive PI label, we derive PI labels with the predictive confidence from a PI label estimation network to quantify the reliability of such graph structure. **(2)** We explore a novel decoupled training approach by leveraging the estimated PI labels for learning a node classification model to perform noise-robust node classification (Section 3.2). We propose to decouple the PI label estimation procedure from training with noisy node labels to prevent corruption on the estimated PI labels. Meanwhile, different from previous works Li et al. (2021b), our `PI-GNN` does not require a clean set of node and label pairs as extra supervision and can simultaneously utilize both the labeled and unlabeled nodes for training, which works well for semi-supervised node classification.

Our main contributions are summarized as follows:

- We propose to train robust GNNs against noisy labels for node classification, which serve as a crucial step towards the reliable deployment of GNNs in complex real-world applications.

- We introduce a novel learning framework to simultaneously estimate and leverage the pairwise interactions, which can be applied on both labeled and unlabeled nodes without extra supervision of clean node labels.

- We demonstrate `PI-GNN` can be effectively used on different datasets, GNN architectures and different noise types and rates, *e.g.*, improving the test accuracy by 5.4% on CiteSeer under a severe label noise.

## 2 Preliminaries

**Graph Neural Networks.** Let $G = (V, E, A)$ be a graph with node feature vectors $X_v$ for $v \in V$ and edge set $E$, where $A$ denotes the adjacency matrix. GNNs use the graph structure and node features $X_v$ to learn a representation vector of a node $h_v$, or the entire graph $h_G$, which usually follow a neighborhood aggregation strategy and iteratively update the representation of a node by aggregating representations of its neighbors. After $k$ iterations of aggregation, a node's representation captures the structural information within its $k$-hop network neighborhood. Formally, the $k$-th layer of a GNN is

$$a_v^{(k)} = \text{AGGREGATE}^{(k)}\left(\left\{h_u^{(k-1)} : u \in \mathcal{N}(v)\right\}\right), h_v^{(k)} = \text{COMBINE}^{(k)}\left(h_v^{(k-1)}, a_v^{(k)}\right), \quad (1)$$

where $h_v^{(k)}$ is the feature vector of node $v$ at the $k$-th layer. $h_v^{(0)} = X_v$. $\mathcal{N}(v)$ denotes the neighboring nodes of $v$. The choices of $\text{AGGREGATE}^{(k)}(\cdot)$ and $\text{COMBINE}^{(k)}(\cdot)$ can be diverse among different GNNs. For

---

[1]There are some additional cases where the conclusion might not hold true, please see Appendix Section O

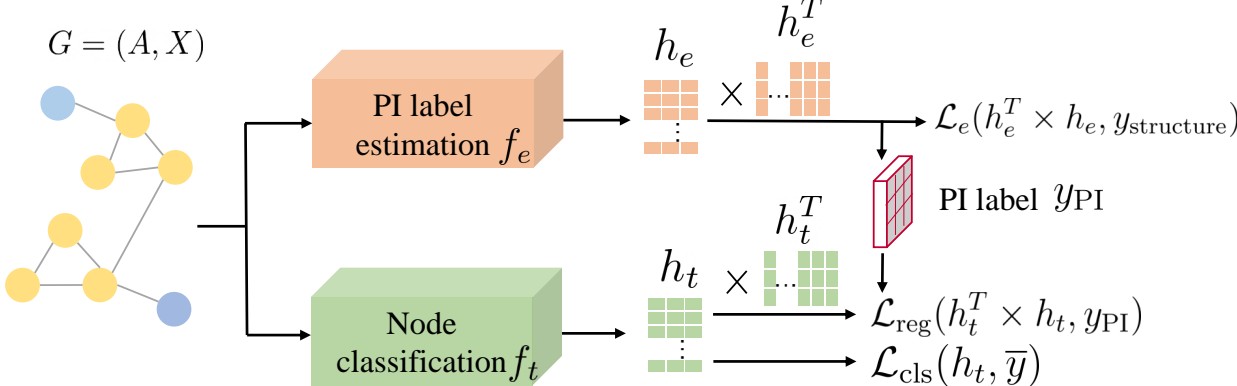

Figure 2: **The framework of our proposed PI-GNN**, which consists of two different branches, *i.e.*, a PI label estimation branch and a node classification branch for noise-robust semi-supervised node classification. The two branches $f_e, f_t$ first estimate the pairwise interactions between each node pair by the graph structure, and then leverage the estimated PI labels for joint training with the node classification task. $\times$ denotes the dot product operation.

example, in GCN Kipf & Welling (2017), the element-wise mean pooling is used, and the AGGREGATE and COMBINE steps are integrated as follows:

$$h_v^{(k)} = \text{ReLU}\left(W \cdot \text{MEAN}\left\{h_u^{(k-1)}, \forall u \in \mathcal{N}(v) \cup \{v\}\right\}\right), \tag{2}$$

where $W$ is a learnable matrix. For node classification, each node $v \in V$ has an associated label $y_v$, the node representation $h_v^{(K)}$ of the final layer is used for prediction.

**Label-noise representation learning for GNNs.** Let $X_v$ be the feature and $y_v$ be the label for node $v$, we deal with a dataset $\mathcal{D} = \{\overline{\mathcal{D}}^{\text{tr}}, \mathcal{D}^{\text{te}}\}$ which consists of training set $\overline{\mathcal{D}}^{\text{tr}} = \{(A, X_v, \overline{y}_v)\}_{v \in V}$ that is drawn from a corrupted distribution $\overline{D} = p(A, X, \overline{Y})$ where $\overline{Y}$ denotes the corrupted label. Let $p(A, X, Y)$ be the non-corrupted joint probability distribution of features $X$ and labels $y$, and $f^*$ be the (Bayes) optimal hypothesis from $X$ to $y$. To approximate $f^*$, the objective requires a hypothesis space $\mathcal{H}$ of hypotheses $f_\theta(\cdot)$ parametrized by $\theta$. A robust algorithm against noisy labels contains the optimization policy to search through $\mathcal{H}$ in order to find $\theta^*$ that corresponds to the optimal function in the hypothesis for $\overline{\mathcal{D}}^{\text{tr}} : f_{\theta^*} \in \mathcal{H}$, and meanwhile is able to assign correct labels for $\mathcal{D}^{\text{te}}$.

## 3 Proposed Approach

In this section, we introduce our proposed PI-GNN, which performs noise-robust semi-supervised node classification by explicitly estimating and leveraging the pairwise interactions on graphs. In what follows, we will first provide a method overview and then illustrate the confidence-aware estimation of the pairwise interactions in PI-GNN (Section 3.1). We introduce the decoupled training strategy for leveraging the estimated pairwise interactions for model regularization (Section 3.2).

**Overview.** Figure 2 demonstrates the overview of PI-GNN, which is composed of two different branches. The confidence-aware PI label estimation branch takes in the graph structure and generates the estimated PI labels for a node pair. We denote it as $f_e$. The node classification branch takes the estimated PI labels and trains a node classification model jointly with an additional regularization objective, which leverages the PI labels to regularize the node embeddings. We denote it as $f_t$.

### 3.1 Confidence-aware PI Label Estimation

Let us suppose certain node class labels in the training set $\overline{\mathcal{D}}^{\text{tr}} = \{(A, X_v, \overline{y}_v)\}_{v \in V}$ are corrupted with noisy labels. Since, ultimately, we are interested in finding a GNN model $f$ parametrized by $\theta$ that minimizes the generalization error on a clean test set $\mathcal{D}^{\text{te}}$, a natural solution is to exploit additional information during training for the learning algorithm to find a robust parameter $\theta^*$ in the hypothesis space $\mathcal{H}$. One

straightforward candidate for such information is leveraging the pairwise interactions between two nodes to perform extra regularization, whose learning objective is shown to hold a much smaller noise rate than that with the noisy class labels (Figure 1).

**Train the PI label estimation model.** In this paper, we propose to estimate the PI labels $y_{\text{PI}} \in \mathbb{R}^{|V| \times |V|}$ between node pairs by learning from the graph structure (Here $|V|$ is the cardinality of the vertex set on the input graph $G$). While a reasonable choice of $y_{\text{PI}}$ is by comparing whether two nodes have the same class label $y$ and assigning those with the same class label a positive PI label, it is *impossible* to obtain such PI labels with noisy class labels $\overline{y}$.

Therefore, we propose to learn from the graph structure $A$ for estimating the PI labels using a PI label estimation model $f_e$ with paprameter $\theta_e$, such as a graph neural network. Specifically, given node embeddings $h$ which is calculated by $h = f_e(A, X, \theta_e)$, assume $i, j \in V$, let $h_i^T \cdot h_j$ be the dot product of two node embeddings, the training objective for the PI label estimation model $\mathcal{L}_e \in \mathbb{R}^{|V| \times |V|}$ is formulated as follows:

$$\mathcal{L}_e\left(h; A, X\right) = \lambda \cdot \mathbb{E}_{(i,j) \in B^-}\left[-\log \frac{1}{1 + \exp^{h_i^T \cdot h_j}}\right] + \mathbb{E}_{(i,j) \in B^+}\left[-\log \frac{\exp^{h_i^T \cdot h_j}}{1 + \exp^{h_i^T \cdot h_j}}\right], \tag{3}$$

where $B^+, B^-$ denote the node pairs that are connected and disconnected, respectively. $\lambda$ is applied for the disconnected node pairs to deal with the sample imbalance problem, which can be calculated according to Kipf & Welling (2016; 2017).

**Estimate the PI labels.** Given a well-trained PI label estimation model $f_e$, we derive the PI labels by taking the predictive confidence as the smoothed PI label, which is calculated as follows:

$$y_{\text{PI}}(i, j) = \frac{\exp^{h_i^T \cdot h_j}}{1 + \exp^{h_i^T \cdot h_j}}. \tag{4}$$

---

**Algorithm 1** `PI-GNN`: Noise-robust Graph Learning by Estimating and Leveraging Pairwise Interactions

---

**Input:** Input graph $G = (V, A, X)$ with noisy training data $\overline{\mathcal{D}}^{\text{tr}} = \{(A, X_v, \bar{y}_v)\}_{v \in V}$, randomly initialized GNNs $f_e$ and $f_t$ with parameter $\theta_e$ and $\theta_t$, weight for regularization loss $\beta$, pretraining epoch $K$ for $f_e$. Total training epoch $N$.
**Output:** Robust GNN $f_t$.
**for** $epoch = 0; epoch < N; epoch + +$ **do**
    **if** $epoch \leq K$ **then**
        Update the parameter $\theta_e$ of the PI label estimation model $f_e$ by Equation 3.
        Set $\beta = 0$ in Equation 5, update the parameter $\theta_t$ of the node classification model $f_t$.
    **else**
        Update the parameter $\theta_e$ of the PI label estimation model $f_e$ by Equation 3.
        Estimate the PI label $y_{\text{PI}}$ by Equation 4 with $f_e$.
        Update the parameter $\theta_t$ of the node classification model $f_t$ by Equation 5.
    **end**
**end**
**return** The node classification model $f_t$.

---

where $y_{\text{PI}}(i, j)$ is proportional to the value of the dot product $h_i^T \cdot h_j$. The estimated PI label measures the predictive confidence by looking at the closeness between the prediction and the graph structure (*i.e.,* node connectivity). If the predictive confidence becomes far away from the binary node connectivity (0 or 1), then the reliability of such predictions is relatively low, which results in a smoothed PI label and vice versa. We show in Section 4.5 the ablations of training with different kinds of PI labels, including training with the PI labels by comparing the clean node labels.

The PI label estimation procedure allows for explicitly exploiting the pairwise interactions between two nodes, resulting in a GNN that is affected less by the noisy class labels. Wu et al. (2021) employed similarity labels

for regularization. However, it transforms the noise transition matrix estimated from the noisy class labels $\overline{y}$ to correct the similarity labels, which is sensitive to the matrix estimation quality. Meanwhile, their approach did not deal with the noise-robust graph learning problem.

### 3.2 Decoupled Noise-Robust Training

Given the well-estimated PI labels, now we discuss how to leverage the PI labels for noise-robust training. One simple solution is to train a single GNN to estimate the PI labels, leverage the estimated PI labels by replacing the binary labels $y_{\text{structure}}$ with the estimated smoothed PI labels and then perform regularization by joint training with the node classification task on that single GNN. However, since the GNN is exposed to the noisy node class labels $\overline{y}$, *the PI label $y_{PI}$ cannot be estimated well if we entangle both the node label prediction and PI label estimation in a single GNN.*

**Decoupling with two branches.** In this paper, as shown in Figure 2, we propose to decouple the PI label estimation and node classification by using two separate GNNs, which are referred as a PI label estimation model $f_e$ and a node classification model $f_t$. The PI label estimation model generates the predictive confidence $y_{\text{PI}}$ by only learning with the PI estimation objective (Equation 3). The node classification model uses the generated PI label $y_{\text{PI}}$ from the PI label estimation model at the same time for model regularization.

**Overall training.** Put them together, we introduce a new noise-robust training objective for node classification against noisy labels on GNNs, leveraging the estimated PI label in Section 3.1. The key idea is to perform the node classification task by the node classification model $f_t$ while regularizing $f_t$ to produce similar embeddings for nodes that have a larger PI label and vice versa. Different from Equation 3 that uses the discrete labels of 0 and 1, we use the smoothed PI label as the learning target. The overall noise-robust training objective for the node classification branch $f_t$ is formulated as:

$$\mathcal{L}_t = \mathcal{L}_{\text{cls}}(f_t(A, X, \theta_t), \overline{y}) + \beta \cdot \mathcal{L}_{\text{reg}}, \tag{5}$$

where $\beta$ is a hyperparameter to balance the node classification loss $\mathcal{L}_{\text{cls}}$ and the regularization loss $\mathcal{L}_{\text{reg}}$ using the estimated PI labels. Concretely, $\mathcal{L}_{\text{reg}}$ is defined as follows:

$$\begin{aligned}
\mathcal{L}_{\text{reg}} = \lambda \cdot \mathbb{E}_{(i,j)\in B^-} &\left[ -(1 - y_{\text{PI}}(i,j)) \cdot \log \frac{1}{1 + \exp^{h_i^T \cdot h_j}} - y_{\text{PI}}(i,j) \cdot \log \frac{\exp^{h_i^T \cdot h_j}}{1 + \exp^{h_i^T \cdot h_j}} \right] \\
+ \mathbb{E}_{(i,j)\in B^+} &\left[ -(1 - y_{\text{PI}}(i,j)) \cdot \log \frac{1}{1 + \exp^{h_i^T \cdot h_j}} - y_{\text{PI}}(i,j) \cdot \log \frac{\exp^{h_i^T \cdot h_j}}{1 + \exp^{h_i^T \cdot h_j}} \right]
\end{aligned} \tag{6}$$

where $y_{\text{PI}}(i,j)$ is defined in Equation 4.

Besides, the PI label estimation model is trained only by the binary classification loss $\mathcal{L}_e$ (Equation 3) and provides the estimated PI labels for model regularization on the node classification model, which is not affected by the noisy class labels. During inference, we discard the PI label estimation model $f_e$ and only use the node classification model $f_t$ for evaluation, which does not affect the inference speed.

Practically, the learning procedure relies heavily on the quality of the PI label estimation by $f_e$. Therefore, in the implementation, we pretrain the PI label estimation model $f_e$ for $K$ epochs (meanwhile we set $\beta$ in Equation 5 to 0 to train the node classification model as well) and then jointly train the two models together by Equations 3 and 5, respectively. Our algorithm is summarized in Algorithm 1.

**Time complexity analysis.** Assume the GNN architecture for the node classification model and the PI label estimation model is GCN, for the node classification model, the time complexity is $\mathcal{O}\left(TL|E|d + TL|V|d^2\right)$ according to Wang et al. (2021) where $d$ is the dimension of the node embedding, $|E|, |V|$ are the number of edges and nodes of the graph and $T, L$ are the number of iterations and the layers. For the PI label estimation model, the complexity is $\mathcal{O}\left(TL|E|d + TL|V|d^2 + T|V|^2 d^2\right)$. For big graph datasets with large $|E|$ and $|V|$, we perform subgraph sampling Hamilton et al. (2017) in the implementation (Section 4.2) to reduce the time complexity.

| Noise type | No Noise | Symmetric Noise | | | | Asymmetric Noise | | | |
|---|---|---|---|---|---|---|---|---|---|
| Noise ratio | 0.0 | 0.2 | 0.4 | 0.6 | 0.8 | 0.2 | 0.4 | 0.6 | 0.8 |
| Cora | | | | | | | | | |
| GCN | **0.804(0.01)** | 0.722(0.03) | 0.613(0.07) | 0.446(0.06) | 0.285(0.07) | 0.703(0.04) | 0.514(0.06) | 0.291(0.04) | 0.161(0.02) |
| PI-GNN w/o pc | 0.781(0.01) | 0.731(0.02) | 0.654(0.05) | 0.510(0.04) | 0.287(0.06) | 0.717(0.04) | 0.563(0.07) | 0.332(0.06) | 0.209(0.06) |
| PI-GNN | 0.780(0.01) | **0.739(0.02)** | **0.664(0.03)** | **0.515(0.03)** | **0.296(0.05)** | **0.723(0.03)** | **0.587(0.07)** | **0.350(0.07)** | **0.232(0.06)** |
| CiteSeer | | | | | | | | | |
| GCN | 0.683(0.01) | 0.603(0.02) | 0.524(0.04) | 0.382(0.04) | 0.230(0.03) | 0.595(0.03) | 0.465(0.05) | 0.281(0.05) | 0.171(0.05) |
| PI-GNN w/o pc | 0.656(0.03) | 0.606(0.03) | 0.526(0.05) | 0.378(0.05) | 0.227(0.04) | 0.588(0.04) | 0.472(0.05) | 0.328(0.03) | 0.235(0.03) |
| PI-GNN | **0.684(0.03)** | **0.642(0.03)** | **0.591(0.03)** | **0.432(0.07)** | **0.245(0.05)** | **0.628(0.03)** | **0.531(0.06)** | **0.353(0.06)** | **0.238(0.06)** |
| PubMed | | | | | | | | | |
| GCN | **0.786(0.01)** | 0.707(0.02) | 0.610(0.06) | 0.462(0.07) | 0.367(0.07) | 0.682(0.05) | 0.524(0.08) | 0.399(0.06) | 0.387(0.07) |
| PI-GNN w/o pc | 0.774(0.00) | 0.723(0.03) | 0.628(0.05) | 0.458(0.07) | 0.370(0.06) | 0.722(0.03) | 0.579(0.07) | 0.412(0.05) | 0.401(0.03) |
| PI-GNN | 0.774(0.00) | **0.724(0.03)** | **0.638(0.04)** | **0.470(0.08)** | **0.379(0.07)** | **0.723(0.03)** | **0.583(0.07)** | **0.425(0.07)** | **0.406(0.04)** |
| WikiCS | | | | | | | | | |
| GCN | **0.703(0.01)** | 0.635(0.03) | 0.558(0.04) | 0.376(0.05) | 0.183(0.05) | 0.608(0.05) | 0.468(0.05) | 0.272(0.05) | 0.129(0.07) |
| PI-GNN w/o pc | 0.676(0.01) | 0.624(0.02) | 0.552(0.05) | 0.396(0.07) | 0.197(0.07) | 0.607(0.03) | 0.470(0.05) | 0.290(0.05) | 0.125(0.05) |
| PI-GNN | 0.676(0.01) | **0.636(0.02)** | **0.562(0.04)** | **0.398(0.07)** | **0.208(0.07)** | **0.610(0.04)** | **0.483(0.05)** | **0.303(0.04)** | **0.135(0.06)** |
| OGB-arxiv | | | | | | | | | |
| GCN | **0.491(0.01)** | 0.461(0.01) | 0.433(0.01) | 0.393(0.03) | 0.278(0.02) | 0.435(0.01) | 0.399(0.01) | 0.059(0.01) | 0.019(0.00) |
| PI-GNN w/o pc | 0.462(0.04) | 0.469(0.08) | 0.445(0.05) | 0.406(0.05) | 0.357(0.10) | 0.445(0.08) | 0.425(0.06) | 0.060(0.02) | 0.021(0.00) |
| PI-GNN | 0.482(0.01) | **0.476(0.04)** | **0.467(0.03)** | **0.418(0.04)** | **0.368(0.09)** | **0.475(0.01)** | **0.461(0.01)** | **0.069(0.01)** | **0.022(0.00)** |

Table 1: Test accuracy on 5 datasets for `PI-GNN` with GCN as the backbone. **Bold** numbers are superior results. Std. is shown in the bracket. w/o pc means that the PI labels are not estimated using the predictive confidence but just the node connectivity.

**Underlying assumption and limitation.** We note that the proposed method `PI-GNN` has a limitation during deployment, which relies on the assumption about the underlying graph structure and their label distribution. Namely, homophilous graphs (*i.e.*, if two nodes are connected, then with high probability, they should have the same node label) are more suited to `PI-GNN` compared to heterophilous graphs (Zhu et al., 2020; Lim et al., 2021; Chien et al., 2022). As a quick verification, we test the performance of `PI-GNN` on several heterophilous datasets in Appendix Section E, where the improvement of `PI-GNN` is less obvious compared to `PI-GNN` on homophilous graphs.

## 4 Experiments and Results

In this section, we present empirical evidence to validate the effectiveness of `PI-GNN` on different datasets with different noise types and ratios.

### 4.1 Experimental setting

**Datasets.** We used five datasets to evaluate `PI-GNN`, including Cora, CiteSeer and PubMed with the default dataset split as in (Kipf & Welling, 2017) and WikiCS dataset (Mernyei & Cangea, 2020) as well as OGB-arxiv dataset (Hu et al., 2020). For WikiCS, we used the first 20 nodes from each class for training and the next 20 nodes for validation. The remaining nodes for each class are used as the test set. For OGB-arxiv, we use the default split. *The dataset statistics are summarized in Appendix Section C.*

Since all datasets are clean, following Patrini et al. (2017), we corrupted these datasets manually by the noise transition matrix $Q_{ij} = \Pr(\overline{y} = j \mid y = i)$ given that noisy $\overline{y}$ is flipped from clean $y$. Assume the matrix $Q$ has two representative structures: (1) Symmetry flipping (van Rooyen et al., 2015); (2) Asymmetric pair flipping: a simulation of fine-grained classification with noisy labels, where labelers may make mistakes only within very similar classes. Note the asymmetric case is much harder than the symmetry case. *Their precise definition is in Appendix Section A.* We tested four different noise rates $\epsilon \in \{0.2, 0.4, 0.6, 0.8\}$ in this paper for two different noise types, which cover lightly and extremely noisy supervision. Note that in the most extreme case, the noise rate 80% for pair flipping means 80% training data have wrong labels that cannot be learned without additional assumptions.

**Implementation details.** We used three different GNN architectures, *i.e.*, GCN, GAT and GraphSAGE, which are implemented by `torch-geometric` [2] (Fey & Lenssen, 2019). All of them have two layers. Specifically, the hidden dimension of GCN, GAT and GraphSAGE is set to 16, 8 and 64. GAT has 8 attention heads in the first layer and 1 head in the second layer. The mean aggregator is used for GraphSAGE. We applied Adam optimizer (Kingma & Ba, 2015) with a learning rate of 0.01 for GCN and GraphSAGE and 0.005 for

---

[2] https://github.com/pyg-team/pytorch_geometric/blob/master/examples

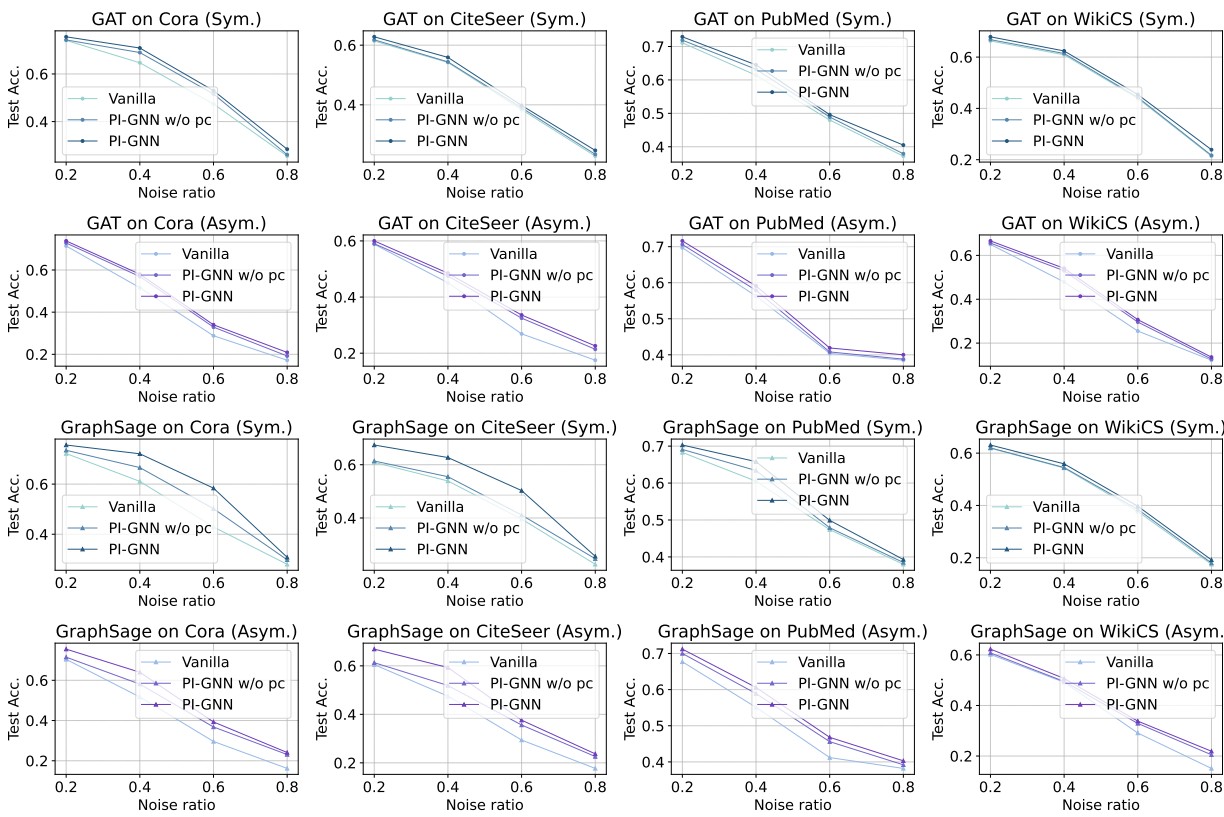

Figure 3: Test accuracy of `PI-GNN` and comparison with `PI-GNN` w/o pc and vanilla GNN on two additional model architectures under different noisy settings.

GAT. The weight decay is set to 5e-4. We trained for 400 epochs on a Tesla P40. The loss weight $\beta$ is set to $|V|^2/(|V|^2 - Q)^2$, where $|V|$ is the number of nodes and $Q$ is the sum of all elements of the preprocessed adjacency matrix. The number of pretraining epochs $K$ is set to 50 and the total epoch $N$ is set to 400. For subgraph sampling, we sampled 15 and 10 neighbors for each node in the 1st and 2nd layer of the GNN and set the batch size to be 1024. We tuned all the hyperparameters on the validation set and reported the node classification accuracy on the clean test set. *Details about the ablation studies on these factors are shown in Section 4.5.* Each experiment is repeated for 10 times with random seeds from 1 to 10. *The training time for our `PI-GNN` and the vanilla GNN model is compared in the Appendix Section I.*

## 4.2 Effectiveness on different datasets

We evaluated the effectiveness of `PI-GNN` on five datasets with different noisy labels and noise rates, which is shown in Table 1 with GCN as the backbone. Specifically, we are interested to observe 1) whether the introduced regularization objective between nodes can improve a vanilla GNN against noisy labels and 2) whether the predictive confidence from the PI label estimation model is beneficial for the test accuracy. Therefore, we compared the accuracy of a vanilla GNN, `PI-GNN` trained with the node connectivity as the PI label (`PI-GNN` w/o pc) and `PI-GNN`.

From Table 1, we made several observations: **Firstly**, the GNN trained with the PI regularization objective is more robust to noisy labels, where both `PI-GNN` w/o pc and `PI-GNN` perform much better than a vanilla GNN. **Secondly**, by estimating the PI labels using the PI label estimation model, the test accuracy is further improved compared to directly using the node connectivity as the PI label. For instance, `PI-GNN` improves the accuracy by 2.3% with the asymmetric noise (noise ratio $\epsilon = 0.8$) on Cora compared to `PI-GNN` w/o pc, which justifies the effectiveness of our design. **Thirdly**, the `PI-GNN` does not help the GNN with the clean

| Noise type | Symmetric Noise | | Asymmetric Noise | |
|---|---|---|---|---|
| Noise ratio | 0.4 | 0.6 | 0.4 | 0.6 |
| | Test dataset: Cora / CiteSeer | | | |
| Decoupling | 0.581(0.06) / 0.518(0.03) | 0.425(0.06) / 0.390(0.03) | 0.541(0.05) / 0.474(0.04) | 0.336(0.03) / 0.323(0.05) |
| GCE | 0.627(0.07) / 0.530(0.03) | 0.447(0.06) / 0.383(0.03) | 0.511(0.05) / 0.468(0.05) | 0.284(0.03) / 0.285(0.05) |
| APL | 0.624(0.08) / 0.522(0.04) | 0.446(0.06) / 0.376(0.04) | 0.507(0.06) / 0.456(0.06) | 0.281(0.03) / 0.281(0.04) |
| Co-teaching | 0.577(0.11) / 0.573(0.07) | 0.376(0.07) / 0.404(0.06) | 0.457(0.10) / 0.462(0.08) | 0.237(0.09) / 0.256(0.08) |
| LPM-1 | 0.542(0.09) / 0.467(0.06) | 0.447(0.07) / 0.395(0.08) | 0.481(0.07) / 0.506(0.08) | 0.318(0.04) / 0.341(0.09) |
| T-Revision | 0.596(0.06) / 0.518(0.03) | 0.425(0.06) / 0.380(0.04) | 0.512(0.06) / 0.457(0.06) | 0.281(0.05) / 0.263(0.05) |
| DivideMix | 0.628(0.06) / 0.515(0.05) | 0.463(0.09) / 0.355(0.05) | 0.428(0.01) / 0.396(0.03) | 0.313(0.03) / 0.282(0.02) |
| PI-GNN (ours) | **0.664(0.03) / 0.591(0.03)** | **0.515(0.03) / 0.432(0.07)** | **0.587(0.07) / 0.531(0.06)** | **0.350(0.07) / 0.353(0.06)** |

Table 2: Comparative results with baselines. **Bold** numbers are superior results. LPM-1 means one extra clean label is used for each class. The result on the left and right of each cell is the classification accuracy of the Cora dataset and CiteSeer dataset, respectively.

node labels, *e.g.*, 80.4% of a vanilla GCN vs. 78.0% of `PI-GNN` on Cora, which illustrates the `PI-GNN` helps to combat noisy supervision rather than inherently improve the node classification with purely clean node labels. *Additional results on heterophilous datasets and with lower noise ratios are in Appendix Sections E and H.*

### 4.3 Performance on different GNN architectures

We evaluated `PI-GNN` on different GNN architectures, *i.e.*, GAT and GraphSAGE. The experiments are conducted on Cora, CiteSeer, PubMed and WikiCS datasets, which are shown in Figure 3. As can be observed, `PI-GNN` performs similarly on GAT and GraphSAGE compared to the results on GCN, where the regularization of PI and the predictive confidence are both beneficial for model generalization even with extremely noisy supervision. Moreover, using the predictive confidence as PI labels is more effective on GraphSAGE. For example, in the Cora dataset, `PI-GNN` improves `PI-GNN` w/o pc by 4.2% and 3.1% on average under symmetric and asymmetric noise, respectively, which is larger than that for GAT and GCN. It may suggest the mean aggregator in GraphSAGE is more susceptible to the sub-optimal PI labels. *We provide significance test on the results of GAT in Appendix Section F.*

### 4.4 Comparison with baselines

In order to further demonstrate the competitive performance of `PI-GNN`, we compared with several powerful baselines for combating noisy labels in literature. For a fair comparison, we used the same GNN architecture, *i.e.*, GCN, and the same overlapping hyperparameters during implementation. The other method-specific hyperparameters are tuned according to the original paper on the validation set. Specifically, we compared with noise-transition matrix-based method, T-revision (Xia et al., 2019), robust loss functions, such as Generalized Cross Entropy (GCE) loss (Zhang & Sabuncu, 2018) and Active Passive Loss (APL) (Ma et al., 2020), optimization-based approaches, such as Co-teaching (Han et al., 2018b), Decoupling (Malach & Shalev-Shwartz, 2017) and DivideMix (Li et al., 2020b). We also compared with Label Propagation and Meta learning (LPM) (Xia et al., 2021), a method that is specifically designed for solving label noise for node classification but uses a small set of clean labels. For illustration, we reported the classification accuracy on Cora and CiteSeer with symmetric and asymmetric noise (noise rate $\epsilon = 0.4, 0.6$) in Table 2.

From Table 2, `PI-GNN` outperforms different baselines with a considerable margin, *e.g.*, improving the classification accuracy by 5.2% on Cora under the symmetric noise ($\epsilon = 0.6$) compared to the best baseline. Moreover, `PI-GNN` is able to outperform LPM-1, which relieves the strong assumption that auxiliary clean node labels are available. *We compare with traditional graph semi-supervised learning approaches and on more datasets in Appendix Section D and Section J, respectively.*

### 4.5 Ablation studies

**The effect of PI labels.** To show the importance of informative PI labels, we tested `PI-GNN` under different PI labels in addition to the one that is estimated by the PI label estimation model (ours): 1) random labels, 2) noisy class label comparison and 3) estimated PI w/o $f_e$ where only one GNN is used for both node classification and PI label estimation. We also compared with the PI labels by clean

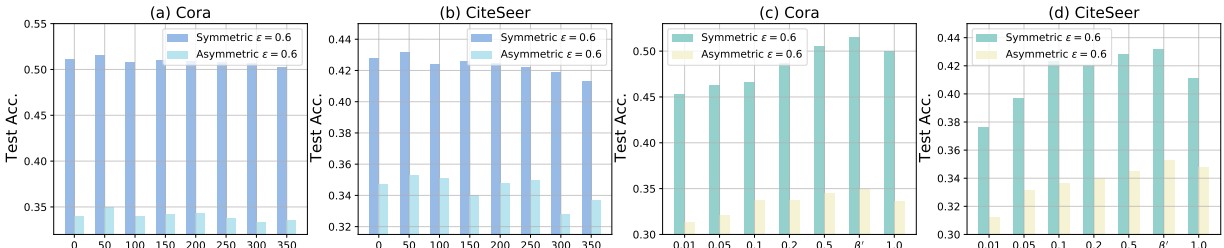

Figure 4: (a)-(b) Performance of `PI-GNN` *w.r.t.* different pretraining epochs on Cora and CiteSeer. x axis denotes the value of the pretraining epochs for the PI label estimation model. (c)-(d) Performance of `PI-GNN` *w.r.t.* the regularization loss weight $\beta$. x axis denotes the value of the loss weight and $\beta'$ is the weight that is aware of the sparsity of the input graph (*cf.* Section 4.1).

class label comparison, which is expected to be the oracle. We used GCN as the backbone and tested on two datasets, *i.e.*, Cora and CiteSeer and different noise types. The results are shown in Table 3.

From Table 3, using PI labels based on the PI label estimation model achieves better performance than using noisy node label comparison. Employing randomly generated PI labels incurs the worst performance because it completely ignores the importance of the pairwise interactions. Moreover, removing the PI label estimation model decreases the test accuracy because the node embeddings are optimized by the noisy class labels as well, which might not be effective for the PI label estimation (32.9% vs. 35.0% for Cora with 60% asymmetric noise). Finally, our `PI-GNN` achieves a similar performance compared to training by clean label comparison (the oracle case), showcasing the effectiveness of our estimated PI labels. Note in our noisy setting, it is impossible to reach the oracle. *More results are in Appendix Section G.*

| Noise Type | Symmetric Noise | Asymmetric Noise |
|---|---|---|
| Cora | | |
| Noise Ratio | 0.6 | 0.6 |
| Random PI label | 0.449(0.05) | 0.295(0.04) |
| Noisy label comparison | 0.471(0.05) | 0.300(0.04) |
| Estimated PI w/o $f_e$ | 0.511(0.03) | 0.329(0.05) |
| Clean label comparison (oracle) | 0.447(0.07) | 0.355(0.05) |
| Estimated PI (ours) | **0.515(0.03)** | **0.350(0.07)** |
| CiteSeer | | |
| Noise Ratio | 0.6 | 0.6 |
| Random PI label | 0.351(0.04) | 0.298(0.04) |
| Noisy label comparison | 0.378(0.04) | 0.312(0.04) |
| Estimated PI w/o $f_e$ | 0.430(0.07) | 0.340(0.05) |
| Clean label comparison (oracle) | 0.447(0.07) | 0.355(0.05) |
| Estimated PI (ours) | **0.432(0.07)** | **0.353(0.06)** |

Table 3: Performance of `PI-GNN` with different PI labels.

**Sensitivity to the pretraining epoch of the PI label estimation model.** We investigated whether the performance of `PI-GNN` is sensitive to the number of pretraining epochs for the PI label estimation model. The experimental results on Cora and CiteSeer with GCN under symmetric and asymmetric noise are shown in Figure 4 (a) and (b). For illustration, we set the noise ratio to be 0.6. As can be observed, pretraining the PI label estimation model for $K$ epochs is effective for improving the generalization on the clean test set. Given a small $K$, the confidence mask is not estimated well which is not helpful to apply it on the node classification model for regularization. Meanwhile, $K$ should not be too large in order to sufficiently regularize the node classification model using Equation 5. $K$ is set to 50 for all the experiments.

**Application of `PI-GNN` on label-noise baselines.** To observe whether `PI-GNN` is able to improve the generalization ability for different label-noise baseline models, we extended three representative approaches, *i.e.*, T-revision (Xia et al., 2019), APL (Ma et al., 2020) and DivideMix (Li et al., 2020b) by adding the PI regularization objective dur-

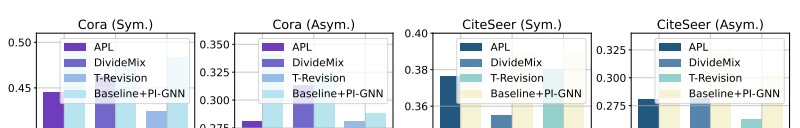

Figure 5: Performance of the `PI-GNN` applied on different label-noise baselines on Cora and CiteSeer. Noise ratio is set to 0.6.

ing training. Specifically, we used the sum of their original loss and the PI regularization loss ($\mathcal{L}_{\text{reg}}$, *cf.* Equation 5) to optimize the GNN. The weight for PI regularization loss is set to the same value as our

approach (*cf.* Section 4.1). We chose GCN as the backbone and reported the test accuracy on Cora and CiteSeer with both symmetric and asymmetric noise ($\epsilon = 0.6$) in Figure 5. As the result shows, `PI-GNN` is orthogonal to those noise-robust baseline models, which is potentially useful for improving their performance without bells and whistles. For instance, The test set accuracy is improved by 4.3% on DivideMix under the asymmetric noise for the CiteSeer dataset and thus demonstrates the universality of our proposed `PI-GNN`.

**Different architectures for two branches.** `PI-GNN` allows for a flexible choice of the architectures for the PI label estimation model and the node classification model, where a light-weight PI label estimation model can help a large node classification model for node classification during training. In what follows, we used three different PI label estimation-node classification model pairs, namely GCN-GAT, GCN-GraphSAGE and GAT-GraphSAGE. The number of parameters for GCN, GAT, GraphSAGE is 0.02, 0.09 and 0.18 M, respectively. The comparison with using the same architectures are shown in Table 4. From Table 4, using a light-weight GNN for the PI label estimation model is able to further improve the clean test

| Noise Type | Sym. Noise | Asym. Noise |
|---|---|---|
| Noise Ratio | 0.6 | 0.6 |
| GAT only | 0.394(0.05) | 0.330(0.04) |
| GCN-GAT | **0.412(0.06)** | **0.339(0.03)** |
| GraphSAGE only | 0.503(0.08) | 0.376(0.08) |
| GCN-GraphSAGE | **0.512(0.09)** | **0.383(0.06)** |
| GraphSAGE only | 0.503(0.08) | 0.376(0.08) |
| GAT-GraphSAGE | **0.516(0.08)** | **0.381(0.06)** |

Table 4: Performance of `PI-GNN` with different architectures for two branches on CiteSeer.

accuracy, which is promising for efficient deployment of `PI-GNN` on real-world graph datasets.

**The effect of regularization weight.** To observe whether the regularization loss weight $\beta$ matters to the model performance, we trained `PI-GNN` with different values of $\beta$, *i.e.*, 0.01, 0.05, 0.1, 0.2, 0.5, 1.0 and compared with the value $\beta' = |V|^2/(|V|^2 - Q)^2$ which is aware of the sparsity of the graph in Figure 4 (c) and (d). We conducted experiments on Cora and CiteSeer with GCN and showed the results with symmetric and asymmetric noise ($\epsilon = 0.6$). From the figure, `PI-GNN` is sensitive to the choice of regularization loss weight $\beta$. On both datasets with different noise types, `PI-GNN` trained with $\beta'$ achieves the best test accuracy, and simultaneously avoids heavy tuning procedure on the validation set.

# 5 Related work

**Graph Neural Networks.** Graph neural networks have been widely used to model the graph-structured data with various architectures, such as graph convolutional network (GCN) (Kipf & Welling, 2017), graph attention network (GAT) (Velickovic et al., 2018), GraphSAGE (Hamilton et al., 2017), Graph Isomorphism Network (GIN) (Xu et al., 2019), Simple Graph Convolution (SGC) (Wu et al., 2019), etc. Common graph analysis tasks, including node classification (Lan et al., 2020), link prediction (Zhang & Chen, 2018), graph classification (Bacciu et al., 2018), graph generation (Liao et al., 2019; Shi et al., 2020), have been widely studied in literature. However, only a few works focused on training robust GNNs against noisy labels, such as by loss correction (NT et al., 2019) for graph classification, sample re-weighting (Xia et al., 2021; Li et al., 2021a) for node classification. None of them exploited explicit PI, which are compared with our `PI-GNN` in Section 4.4. Bui et al. (2017); Stretcu et al. (2019); Ng et al. (2018); Qu et al. (2019); Ma et al. (2019) utilized graph structures for semi-supervised learning but with clean labels. Moreover, they did not further process the graph structure while `PI-GNN` utilizes the graph structure and introduces a new PI label estimation procedure during training. Jiang et al. (2019); Yu et al. (2020); Zheng et al. (2020); Chen et al. (2020b); Kim & Oh (2021); Fatemi et al. (2021) iteratively refined graph structure during training for missing edge prediction or error edge detection while `PI-GNN` does not change the input graph. Luo et al. (2021) proposed a parameterized topological denoising network to improve the robustness and generalization performance of GNNs by learning to *drop task-irrelevant edges*. The main difference is that `PI-GNN` deals with the situation where the node labels are corrupted while they deal with the noisy edges.

Zhao et al. (2020); Stadler et al. (2021); Wu et al. (2023) proposed to use uncertainty estimation and out-of-distribution detection techniques (Du et al., 2022c;b;a; Tao et al., 2023; Yang et al., 2021; Bai et al., 2023; Sun et al., 2021; Sun & Li, 2022; Sun et al., 2022; Ming et al., 2022a;b; 2023; Wei et al., 2022; Yang et al., 2022) for detecting out-of-distribution or noisy node samples on graphs, where our goal is to correctly classify the noisy nodes.

**Neural networks with noisy labels.** Methods for neural networks against noisy labels can be roughly categorized into three types (Han et al., 2020b), *i.e.*, approaches from the perspective of data (van Rooyen & Williamson, 2017), learning objective (Reed et al., 2015; Miyato et al., 2019) and optimization (Arpit et al., 2017). Methods based on data mainly built the noise transition matrix to explore the data relationship between clean and noisy label by an adaptation layer (Sukhbaatar et al., 2015), loss correction (Patrini et al., 2017) and prior knowledge (Han et al., 2018a). Methods based on objective modified the learning objective by regularization (Han et al., 2020a), reweighting (Liu & Tao, 2016; Wang et al., 2017) and loss redesign (Thulasidasan et al., 2019). Methods based on optimization mainly changed the optimization policy, such as by memorization effect (Jiang et al., 2018), self-training (Ren et al., 2018) and co-training (Yu et al., 2019). Wu et al. (2021) proposed to use the similarity loss for noisy labels of images but it relied on the noisy transition matrix, which is sensitive to the matrix estimation quality and cannot use the graph structure for regularization. In this paper, we extend several approaches from each category to compare with `PI-GNN` in Section 4.4.

## 6    Conclusion

In this paper, we proposed `PI-GNN`, a simple but effective learning paradigm for helping the GNN to generalize well with noisy supervision. Our key idea is to leverage the pairwise interactions between nodes to explicitly regularize the similarity of those node embeddings during training. In order to perform noise-robust node classification, we introduce a new learning framework to adaptively estimate and leverage the pairwise interactions for model regularization. We conducted extensive experiments to demonstrate that `PI-GNN` can train GNNs robustly under extremely noisy supervision, which serves as a crucial step towards the reliable deployment of GNNs in complex real-world applications. We hope our work inspires future research on noise-robust graph learning, such as proposing novel pairwise approaches from the algorithmic perspective and constructing real-world noisy graph datasets for a more comprehensive and practical empirical evaluation.

## 7    Acknowledgement

BH was supported by Tencent AI Lab Rhino-Bird Gift Fund. The authors would also like to thank TMLR reviewers for the helpful suggestions and feedback.

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

# Noise-robust Graph Learning by Estimating and Leveraging Pairwise Interactions
# (Appendix)

## A  Definition of noise

The definition of transition matrix $Q$ is as follows. $n$ is number of the class.

Asymmetric pair flipping:

$$Q = \begin{bmatrix} 1-\epsilon & \epsilon & 0 & \dots & 0 \\ 0 & 1-\epsilon & \epsilon & & 0 \\ \vdots & & \ddots & \ddots & \vdots \\ 0 & & & 1-\epsilon & \epsilon \\ \epsilon & 0 & \dots & 0 & 1-\epsilon \end{bmatrix}, \tag{7}$$

Symmetry flipping:

$$Q = \begin{bmatrix} 1-\epsilon & \frac{\epsilon}{n-1} & \dots & \frac{\epsilon}{n-1} & \frac{\epsilon}{n-1} \\ \frac{\epsilon}{n-1} & 1-\epsilon & \frac{\epsilon}{n-1} & \dots & \frac{\epsilon}{n-1} \\ \vdots & & \ddots & & \vdots \\ \frac{\epsilon}{n-1} & \dots & \frac{\epsilon}{n-1} & 1-\epsilon & \frac{\epsilon}{n-1} \\ \frac{\epsilon}{n-1} & \frac{\epsilon}{n-1} & \dots & \frac{\epsilon}{n-1} & 1-\epsilon \end{bmatrix}. \tag{8}$$

## B  Software and hardware

We run all experiments with Python 3.8.5 and PyTorch 1.7.0, using NVIDIA TESLA P40 GPUs.

## C  Dataset Details

Here we provide the details of graph datasets for node classification.

| Dataset | #Nodes | #Edges | #Classes |
|---------|--------|--------|----------|
| Cora | 2,485 | 5,069 | 7 |
| CiteSeer | 2,110 | 3,668 | 6 |
| PubMed | 19,717 | 44,324 | 3 |
| WikiCS | 11,701 | 216,123 | 10 |
| OGB-arxiv | 169,343 | 1,166,243 | 40 |

Table 5: Statistics of the datasets.

## D  Comparison with traditional graph semi-supervised learning based approaches.

For the comparison with the traditional semi-supervised graph embedding methods, we follow the same experimental setting and compare with ICA (Lu & Getoor, 2003), Planetoid (Yang et al., 2016) and Label Propagation (LP) (Zhu & Ghahramani, 2002) on Cora as follows in Table 6. The result shows the advantage of `PI-GNN` across different noise ratios.

## E  Experimental results on heterophilous datasets

We perform extra experiments on heterophilous datasets (Ma et al., 2021). The results are demonstrated in the Table 7. It shows that `PI-GNN` is still able to outperform the vanilla one except for one case in Chameleon

| Noise type | Symmetric Noise | | | | | Asymmetric Noise | | | |
|---|---|---|---|---|---|---|---|---|---|
| Noise ratio | 0.0 | 0.2 | 0.4 | 0.6 | 0.8 | 0.2 | 0.4 | 0.6 | 0.8 |
| ICA | 0.729(0.01) | 0.609(0.01) | 0.523(0.04) | 0.394(0.00) | 0.159(0.00) | 0.549(0.00) | 0.453(0.00) | 0.284(0.01) | 0.127(0.01) |
| LP | 0.603(0.00) | 0.506(0.02) | 0.417(0.03) | 0.297(0.03) | 0.170(0.03) | 0.513(0.03) | 0.391(0.04) | 0.238(0.03) | 0.141(0.02) |
| Planetoid | 0.739(0.01) | 0.639(0.03) | 0.527(0.04) | 0.379(0.05) | 0.265(0.06) | 0.627(0.03) | 0.441(0.04) | 0.271(0.06) | 0.210(0.09) |
| PI-GNN | **0.780(0.01)** | **0.739(0.02)** | **0.664(0.03)** | **0.515(0.03)** | **0.296(0.05)** | **0.723(0.03)** | **0.587(0.07)** | **0.350(0.07)** | **0.232(0.06)** |

Table 6: Comparison with more baselines on Cora Dataset.

dataset. Meanwhile, the improvement is somewhat smaller, which implies PI-GNN may be more effective on homophilous datasets.

| Noise type | Symmetric | | Asymmetric | |
|---|---|---|---|---|
| | Actor | | | |
| Noise Ratio | 0.4 | 0.6 | 0.4 | 0.6 |
| GCN | 0.209(0.02) | 0.208(0.02) | 0.198(0.03) | 0.199(0.02) |
| PI-GNN w/o pc | 0.216(0.01) | 0.210(0.02) | 0.201(0.02) | **0.202(0.02)** |
| PI-GNN | **0.218(0.02)** | **0.213(0.02)** | **0.204(0.02)** | 0.200(0.02) |
| | Chameleon | | | |
| GCN | 0.251(0.03) | 0.246(0.03) | **0.245(0.04)** | 0.228(0.03) |
| PI-GNN w/o pc | 0.264(0.03) | 0.249(0.03) | 0.242(0.05) | 0.229(0.04) |
| PI-GNN | **0.269(0.02)** | **0.251(0.03)** | 0.239(0.05) | **0.237(0.04)** |

Table 7: Experimental results on heterophilous datasets.

# F    Significance test results

We perform significance tests to verify whether PI-GNN outperforms the vanilla GNN model significantly using double-sided T-test in Table 8. We use python package "scipy.stats.ttest1samp" and report the average results over 10 different runs as follows. PI-GNN is better than GAT because the absolute value of the t-statistic is relatively large and the p-value is small.

| Method | Setting | \|T-statistic\| | p-value |
|---|---|---|---|
| | Cora | | |
| GAT vs. PI-GNN | Symmetric Noise-0.8 | 4.78 | 0.001 |
| | Asymmetric Noise-0.8 | 3.42 | 0.008 |
| | CiteSeer | | |
| GAT vs. PI-GNN | Symmetric Noise-0.8 | 2.09 | 0.060 |
| | Asymmetric Noise-0.8 | 4.63 | 0.001 |

Table 8: Statistical significance tests.

# G    Experimental results on using clean label comparison

To observe the node classification results by training with PI labels from clean label comparison (which are obtained by comparing the clean class labels for two nodes), we did experiments on Cora, CiteSeer and PubMed with GCN and a noise ratio of 0.4 and 0.6. The results are shown in the Table 9. In most cases, clean label comparison can help the PI-GNN to combat noisy labels except for some challenging cases with asymmetric noise. One reason may be the inherent noise exists in clean node labels for Cora, where we cannot obtain perfectly clean PI label.

# H    Experimental results with lower noise ratios

For the PI-GNN under lower noise ratios, we empirically verify its effectiveness on Cora and CiteSeer with the noise ratio of 0.1, which is shown in Table 10.

| Noise type | Symmetric | | Asymmetric | |
|---|---|---|---|---|
| | Cora | | | |
| Noise Ratio | 0.4 | 0.6 | 0.4 | 0.6 |
| PI-GNN | 0.664(0.03) | 0.515(0.03) | 0.587(0.07) | **0.350(0.07)** |
| Clean PI-GNN | **0.671(0.03)** | **0.523(0.03)** | **0.589(0.07)** | 0.341(0.07) |
| | CiteSeer | | | |
| PI-GNN | 0.591(0.03) | 0.432(0.07) | 0.531(0.06) | 0.353(0.06) |
| Clean PI-GNN | **0.605(0.04)** | **0.447(0.07)** | **0.536(0.05)** | **0.355(0.05)** |
| | PubMed | | | |
| PI-GNN | 0.638(0.04) | 0.470(0.08) | 0.583(0.07) | 0.425(0.07) |
| Clean PI-GNN | **0.640(0.02)** | **0.485(0.07)** | **0.590(0.07)** | **0.429(0.07)** |

Table 9: Experimental results on using clean label comparison. Clean PI-GNN means the PI-GNN is trained with the PI labels from clean label comparison.

| Noise type | Symmetric | Asymmetric |
|---|---|---|
| | Cora | |
| Noise Ratio | 0.1 | 0.1 |
| GCN | 0.766(0.03) | 0.762(0.04) |
| PI-GNN w/o pc | 0.769(0.03) | 0.763(0.03) |
| PI-GNN | **0.772(0.02)** | **0.768(0.03)** |
| | CiteSeer | |
| Noise Ratio | 0.1 | 0.1 |
| GCN | 0.642(0.03) | 0.618(0.05) |
| PI-GNN w/o pc | 0.648(0.04) | 0.633(0.02) |
| PI-GNN | **0.659(0.03)** | **0.658(0.05)** |

Table 10: Experimental results with lower noise ratios.

## I  Comparison of the training time

We compare the training time of PI-GNN and the vanilla GNN as follows. We observe that using dual GNNs does not incur much higher computational cost because the two GNNs run in parallel rather than in sequence on the GPU and the major time-consuming part is for data loading and transforms rather than forward/backward pass. Additionally, PI-GNN does not incur more inference cost than a vanilla model.

| Dataset | Cora | CiteSeer | PubMed | WikiCS | OGB |
|---|---|---|---|---|---|
| Time-GCN(s) | 25.79 | 26.98 | 140.53 | 71.12 | 3930.14 |
| Time-PI-GNN (s) | 26.46 | 30.33 | 160.29 | 81.95 | 4117.00 |

## J  Additional comparison with baselines

In addition to Table 2 in the main paper, we provide the comparison on PubMed and WikiCS with baselines to further demonstrate the effectiveness of our PI-GNN, which is shown as follows.

| Noise type | Symmetric Noise(0.4) | Asymmetric Noise(0.4) |
|---|---|---|
| | Test dataset: PubMed / WikiCS | |
| Decoupling | 0.627(0.05) / 0.555(0.05) | 0.578(0.06) / 0.465(0.05) |
| GCE | 0.604(0.06) / 0.536(0.06) | 0.524(0.08) / 0.456(0.06) |
| APL | 0.606(0.06) / 0.526(0.09) | 0.524(0.08) / 0.336(0.11) |
| Co-teaching | 0.523(0.06) / 0.337(0.10) | 0.433(0.11) / 0.262(0.08) |
| LPM-1 | 0.634(0.06) / 0.554(0.04) | 0.570(0.09) / 0.401(0.04) |
| T-Revision | 0.603(0.04) / 0.542(0.07) | 0.554(0.08) / 0.478(0.10) |
| DivideMix | 0.543(0.08) / 0.419(0.08) | 0.566(0.07) / 0.169(0.08) |
| PI-GNN (ours) | **0.638(0.04) / 0.562(0.04)** | **0.583(0.07) / 0.483(0.05)** |

Table 11: Additional comparison with baselines on two different datasets. We use the GCN as the graph neural network backbone.

## K   Sensitivity to the model initialization

We provide the sensitivity analysis for PI-GNN on different model initializations in Table 12. The results demonstrate that the performance is not sensitive to different model initializations, where the biggest gap among all the model initializations is smaller than 3%.

| Noise type | Symmetric Noise | Asymmetric Noise |
|---|---|---|
| Noise ratio | 0.6 | 0.6 |
| | Test dataset: Cora / CiteSeer | |
| Uniform Initialization | 0.514(0.05) / 0.443(0.05) | 0.364(0.02) / 0.321(0.04) |
| Normal Initialization | 0.548(0.03) / 0.426(0.05) | 0.361(0.05) / 0.345(0.01) |
| Constant Initialization | 0.504(0.01) / 0.413(0.04) | 0.339(0.03) / 0.341(0.04) |
| Kaiming Initialization | 0.503(0.04) / 0.429(0.03) | 0.333(0.02) / 0.372(0.09) |
| Glorot Initialization (Ours) | 0.515(0.03) / 0.432(0.07) | 0.350(0.07) / 0.353(0.06) |

Table 12: Sensitivity to the model initialization. Model architecture is the GCN.

## L   Results on a larger graph dataset

We evaluate our proposed PI-GNN on an even larger OGB-products dataset, which has 2,449,029 nodes with 61,859,140 edges and thus is much larger than the OGB-arxiv dataset (169,343 nodes and 1,166,243 edges) used in Table 1. The results are shown in Table 13, where our PI-GNN can still demonstrate promise compared to the vanilla GCN model and PI-GNN w/o predictive confidence.

| Noise type | No Noise | Symmetric Noise | | | | Asymmetric Noise | | | |
|---|---|---|---|---|---|---|---|---|---|
| | | OGB-products | | | | | | | |
| Noise ratio | 0.0 | 0.2 | 0.4 | 0.6 | 0.8 | 0.2 | 0.4 | 0.6 | 0.8 |
| GCN | **0.732(0.04)** | 0.709(0.01) | 0.661(0.09) | 0.612(0.07) | 0.471(0.06) | 0.689(0.03) | 0.635(0.03) | 0.102(0.03) | 0.053(0.01) |
| PI-GNN w/o pc | 0.711(0.03) | 0.721(0.04) | 0.669(0.03) | 0.631(0.06) | 0.487(0.09) | 0.700(0.09) | 0.641(0.04) | 0.136(0.01) | 0.110(0.04) |
| PI-GNN | 0.727(0.05) | **0.738(0.06)** | **0.677(0.06)** | **0.658(0.03)** | **0.506(0.05)** | **0.719(0.06)** | **0.667(0.03)** | **0.196(0.06)** | **0.153(0.04)** |

Table 13: Test accuracy on the OGB-products dataset for PI-GNN with GCN as the backbone. **Bold** numbers are superior results. Std. is shown in the bracket. w/o pc means that the PI labels are not estimated using the predictive confidence but just the node connectivity.

## M   Results on using low-rank approximation for PI estimation

We estimate the PI labels by performing SVD on the input graph and using the low-rank representations as the estimation results. Suppose the rank of the representation is $r$, we tested the node classification performance under different values of $r$ (i.e., 1, 50, 100, 200, 400, 600, 1000). Note that we use GCN and

Cora as the GNN architecture and the dataset, respectively. During the low-rank approximation, we force the values of the smoothed reconstruction to be greater than 0 and less than 1 by value clipping. The results are updated in Table 14.

| Noise type | Symmetric Noise | | | | Asymmetric Noise | | | |
|---|---|---|---|---|---|---|---|---|
| Noise ratio | 0.2 | 0.4 | 0.6 | 0.8 | 0.2 | 0.4 | 0.6 | 0.8 |
| GCN | 0.722(0.03) | 0.613(0.07) | 0.446(0.06) | 0.285(0.07) | 0.703(0.04) | 0.514(0.06) | 0.291(0.04) | 0.161(0.02) |
| Low-rank approximation-1 | 0.693(0.01) | 0.587(0.03) | 0.418(0.05) | 0.272(0.03) | 0.688(0.06) | 0.483(0.04) | 0.269(0.02) | 0.110(0.05) |
| Low-rank approximation-50 | 0.698(0.04) | 0.593(0.03) | 0.436(0.03) | 0.269(0.01) | 0.679(0.04) | 0.502(0.06) | 0.271(0.08) | 0.132(0.08) |
| Low-rank approximation-100 | 0.712(0.09) | 0.603(0.05) | 0.427(0.06) | 0.277(0.05) | 0.673(0.01) | 0.515(0.06) | 0.286(0.04) | 0.149(0.02) |
| Low-rank approximation-200 | 0.732(0.04) | 0.607(0.04) | 0.474(0.04) | 0.282(0.03) | 0.694(0.05) | 0.547(0.06) | 0.294(0.01) | 0.173(0.08) |
| Low-rank approximation-400 | 0.734(0.01) | 0.611(0.09) | 0.496(0.05) | **0.300(0.07)** | 0.686(0.01) | 0.563(0.02) | 0.319(0.00) | 0.183(0.05) |
| Low-rank approximation-600 | 0.726(0.02) | 0.626(0.02) | 0.476(0.03) | 0.298(0.03) | 0.695(0.04) | 0.534(0.07) | 0.321(0.03) | 0.201(0.03) |
| Low-rank approximation-1000 | 0.704(0.05) | 0.610(0.08) | 0.455(0.04) | 0.256(0.07) | 0.668(0.02) | 0.508(0.05) | 0.346(0.07) | 0.188(0.09) |
| PI-GNN | **0.739(0.02)** | **0.664(0.03)** | **0.515(0.03)** | 0.296(0.05) | **0.723(0.03)** | **0.587(0.07)** | **0.350(0.07)** | **0.232(0.06)** |

Table 14: Test accuracy on using low-rank approximation as the PI labels.

$r = 400$ roughly achieves the best performance but still cannot outperform our PI-GNN. The reason might be that low-rank approximation is mainly designed for matrix compression, denoising and completion (https://web.stanford.edu/class/cs168/l/l9.pdf), which cannot capture the uncertainty of the PI labels in essence as in PI-GNN. Moreover, we observe that during low-rank approximation, the values of each node pair will quickly goes from negative values to values larger than 1, which is not suitable to be the PI labels for training. Although we directly adopt the clipping approach to force the values to be in the range of 0 and 1, additional curated designs might be more beneficial. Finally, the rank $r$ is an important hyperparameter to tune in the low-rank approximation approaches, which requires extra manual tuning compared to our PI-GNN.

## N   Discusssion on a different training strategy

In Table 15 and 16, we test the performance of PI-GNN using a different training scheme, where we firstly pretrain the PI label estimation network $f_e$ for 400 epochs. Then we directly apply the pretrained $f_e$ to output the estimated PI labels for training $f_t$. We show the results on Cora and CiteSeer datasets as follows:

| Noise type | No Noise | Symmetric Noise | | | | Asymmetric Noise | | | |
|---|---|---|---|---|---|---|---|---|---|
| | | OGB-products | | | | | | | |
| Noise ratio | 0.0 | 0.2 | 0.4 | 0.6 | 0.8 | 0.2 | 0.4 | 0.6 | 0.8 |
| PI-GNN (Pretrain) | 0.772(0.01) | 0.733(0.03) | **0.672(0.04)** | 0.508(0.06) | **0.319(0.04)** | **0.728(0.04)** | 0.569(0.04) | 0.339(0.06) | **0.245(0.06)** |
| PI-GNN (Ours) | **0.780(0.01)** | **0.739(0.02)** | 0.664(0.03) | **0.515(0.03)** | 0.296(0.05) | 0.723(0.03) | **0.587(0.07)** | **0.350(0.07)** | 0.232(0.06) |

Table 15: Test accuracy on Cora using a different training scheme.

| Noise type | No Noise | Symmetric Noise | | | | Asymmetric Noise | | | |
|---|---|---|---|---|---|---|---|---|---|
| | | OGB-products | | | | | | | |
| Noise ratio | 0.0 | 0.2 | 0.4 | 0.6 | 0.8 | 0.2 | 0.4 | 0.6 | 0.8 |
| PI-GNN (Pretrain) | **0.693(0.01)** | 0.631(0.03) | **0.606(0.04)** | **0.452(0.04)** | 0.229(0.05) | 0.604(0.04) | 0.511(0.04) | 0.338(0.02) | **0.242(0.06)** |
| PI-GNN (Ours) | 0.684(0.03) | **0.642(0.03)** | 0.591(0.03) | 0.432(0.07) | **0.245(0.05)** | **0.628(0.03)** | **0.531(0.06)** | **0.353(0.06)** | 0.238(0.06) |

Table 16: Test accuracy on CiteSeer using a different training scheme.

where we can observe that firstly pretraining the PI label estimation network and freezing it during training the node classification model achieves a similar classification performance compared to our training scheme (*c.f.* Algorithm 1).

## O   Additional cases of the teaser example

We note that there are additional cases that the conclusion of the teaser example might not hold. For instance, four nodes have the clean labels as $x1 - 0, x2 - 0, x3 - 0, x4 - 0$, after noise corruption, the node labels are changed to $x1 - 1, x2 - 0, x3 - 0, x4 - 0$. Therefore, the noise ratio of the node labels is 25% while that of the PI labels is 37.5%. We provide the comparison of noise ratios between PI labels and node labels on real-world datasets in Section P in order to verify the validity of our proposed PI-GNN.

## P    Noise ratio comparison between PI labels and node labels on real-world datasets

For evaluating the noise ratio of the PI labels on real-world graph datasets, we compare the noise ratio of the PI labels and node labels on Cora and CiteSeer datasets. The results are shown as follows:

| Cora | | | | |
|---|---|---|---|---|
| Noise ratio of the node labels (Symmetric) | 0.2 | 0.4 | 0.6 | 0.8 |
| Noise ratio of the PI labels | 0.008 | 0.021 | 0.031 | 0.036 |
| Noise ratio of the node labels (Asymmetric) | 0.2 | 0.4 | 0.6 | 0.8 |
| Noise ratio of the PI labels | 0.012 | 0.021 | 0.029 | 0.035 |
| CiteSeer | | | | |
| Noise ratio of the node labels (Symmetric) | 0.2 | 0.4 | 0.6 | 0.8 |
| Noise ratio of the PI labels | 0.009 | 0.017 | 0.024 | 0.034 |
| Noise ratio of the node labels (Asymmetric) | 0.2 | 0.4 | 0.6 | 0.8 |
| Noise ratio of the PI labels | 0.009 | 0.016 | 0.025 | 0.031 |

Table 17: Noise ratio comparison between PI labels and node labels on real-world datasets

From the above table, we can observe the noise ratio of the PI labels is indeed small on the real graph datasets, which is able to justify the intuition of our `PI-GNN`.

## Q    Subgraph sampling on smaller graphs

We test the performance of `PI-GNN` (GCN as the backbone) after using subgraph sampling on smaller graphs, i.e., Cora and CiteSeer datasets. Specifically, we sample 15 and 10 neighbors for each node in the 1st and 2nd layer of the GNN and set the batch size to 128. The results are shown as follows:

| Noise type | No Noise | Symmetric Noise | | | | Asymmetric Noise | | | |
|---|---|---|---|---|---|---|---|---|---|
| | | OGB-products | | | | | | | |
| Noise ratio | 0.0 | 0.2 | 0.4 | 0.6 | 0.8 | 0.2 | 0.4 | 0.6 | 0.8 |
| PI-GNN (Subgraph sampling) | 0.771(0.01) | 0.728(0.03) | 0.651(0.03) | **0.523(0.05)** | **0.311(0.02)** | 0.708(0.03) | 0.573(0.07) | 0.349(0.01) | **0.246(0.07)** |
| PI-GNN (Ours) | **0.780(0.01)** | **0.739(0.02)** | **0.664(0.03)** | 0.515(0.03) | 0.296(0.05) | **0.723(0.03)** | **0.587(0.07)** | **0.350(0.07)** | 0.232(0.06) |

Table 18: Test accuracy on Cora using subgraph sampling.

| Noise type | No Noise | Symmetric Noise | | | | Asymmetric Noise | | | |
|---|---|---|---|---|---|---|---|---|---|
| | | OGB-products | | | | | | | |
| Noise ratio | 0.0 | 0.2 | 0.4 | 0.6 | 0.8 | 0.2 | 0.4 | 0.6 | 0.8 |
| PI-GNN (Subgraph sampling) | 0.674(0.01) | 0.639(0.04) | 0.579(0.07) | 0.430(0.09) | **0.258(0.05)** | 0.618(0.04) | 0.528(0.04) | 0.348(0.03) | **0.250(0.08)** |
| PI-GNN (Ours) | **0.684(0.03)** | **0.642(0.03)** | **0.591(0.03)** | **0.432(0.07)** | 0.245(0.05) | **0.628(0.03)** | **0.531(0.06)** | **0.353(0.06)** | 0.238(0.06) |

Table 19: Test accuracy on CiteSeer using subgraph sampling.

where we can observe that applying subgraph sampling is less effective than using the entire graph as the input on smaller noise ratios.

## R    Comparison with baselines on OGB-arxiv

In Table 20, we implement all the baselines on OGB-arxiv dataset and compare them with `PI-GNN` using subgraph sampling.

where we observe `PI-GNN` can still outperform all the baselines in most cases.

| Noise type | Symmetric Noise | | Asymmetric Noise | |
|---|---|---|---|---|
| Noise ratio | 0.4 | 0.6 | 0.4 | 0.6 |
| | Test dataset: OGB-arxiv | | | |
| Decoupling | 0.385(0.09) | 0.347(0.05) | 0.411(0.04) | 0.029(0.01) |
| GCE | 0.451(0.05) | 0.407(0.02) | 0.391(0.06) | 0.057(0.03) |
| APL | 0.412(0.05) | 0.375(0.05) | 0.399(0.06) | 0.062(0.01) |
| Co-teaching | 0.461(0.04) | 0.403(0.04) | 0.410(0.05) | 0.038(0.01) |
| LPM-1 | 0.450(0.01) | 0.397(0.03) | 0.439(0.06) | 0.056(0.01) |
| T-Revision | 0.417(0.04) | 0.409(0.05) | 0.427(0.05) | **0.071(0.04)** |
| DivideMix | 0.448(0.01) | 0.403(0.05) | 0.438(0.05) | 0.041(0.02) |
| PI-GNN (ours) | **0.467(0.03)** | **0.418(0.04)** | **0.461(0.01)** | 0.069(0.01) |

Table 20: Comparative results with baselines on OGB-arxiv.

