# OpenReview forum: "Noise-robust Graph Learning by Estimating and Leveraging Pairwise Interactions"
_TMLR — Accepted by TMLR_

### Review · Reviewer_kfuH · 2023-02-16

**Summary Of Contributions:**

This paper proposes to use pairwise interactions for semi-supervised node classification with noisy node labels. Experimental results show its effectiveness on some graph datasets.

**Audience:**

Yes

**Broader Impact Concerns:**

So far as I can tell, there is no broader impact concern.

**Claims And Evidence:**

Yes

**Requested Changes:**

See my comments above.

**Strengths And Weaknesses:**

Strength
S1. To the best of my knowledge, estimating pair-wise interactions for noise-robust node classification is novel.
S2. The proposed approach is easy to understand and follow.

Weakness
W1. (Major) The motivation of considering PI is unclear. On the first place, contrary to authors’ claims, noise rate for the PI labels actually might be larger than that of the pointwise noisy class labels (a simple example: x1-0,x2-0,x3-0,x4-0 -> x1-1,x2-0,x3-0, x4-0). While it might be that noise rates for the PI labels are usually small in practice, the authors did not verify it in real-world datasets. Additionally, it is also confusing why smaller PI label noise rate implies that it is easier to estimate PI. In fact, there are more possible pairwise interactions (|V|^2) than pointwise labels (|V|), which might indicate accurately predicting them requires larger model capacity.

W2. (Major) The quadratic computational cost w.r.t. number of nodes is very high. While the authors suggest using subgraph sampling for large graphs, whether such approach will decrease the testing accuracy is unknown. I would suggest the authors to conduct additional experiments on smaller datasets to verify this point and also compare its scalability with other baselines. In addition, table 2 only reports comparison with baselines on two small datasets. I wonder if PI-GNN can still outperform baselines on large datasets (e.g., ogb datasets), after using subgraph sampling.

W3. In experiments, noise is manually generated, which might not be able to reflect the true noise distribution in real-world scenarios.

W4. In figure 1, ’Noise ratio’ is incorrectly placed: 50% should be placed in the pointwise noisy ratio part (uppper panel), and 25% should be placed in the PI noisy ratio part (bottom panel).

W5. Lack of discussions on some closely-related topics such as out-of-distribution node detection and uncertainty prediction [1-3] whose target is to detect out-of-distribution or noisy samples.

[1] Uncertainty aware semi-supervised learning on graph data. in NeurIPS 2020
[2] Graph posterior network: Bayesian predictive uncertainty for node classification. in NeurIPS 2021
[3] Energy-based Out-of-Distribution Detection for Graph Neural Networks. in ICLR 2023

---

> ### Author Response · Authors · 2023-03-11
> **Response to Reviewer kfuH-part I**
>
> We are glad that the reviewer finds our work novel, is easy to understand and follow, and with effective experimental results. We also thank you for the constructive comments and suggestions, which we address below:
>
> **A1. Motivation of using PI**
>
> - Thanks for capturing the blind spot of our teaser example in the original submission! We admit that in the provide example from the reviewer, the noise ratio of the pairwise interactions is greater than that of the pointwise node labels. We have updated the description of the teaser example in the updated submission to clarify this blind spot (c.f. **Section 1, paragraph 3** and **Appendix Section O**)!
>
>     For evaluating the noise ratio of the PI labels on real-world graph datasets, as suggested, we compared the noise ratio of the PI labels and node labels on Cora and CiteSeer datasets. The results are shown in **Appendix Section P** of the updated submission, where we can observe the noise ratio of the PI labels is indeed smaller on the real graph datasets. We also show the new table below:
>
>     For Cora,
>     |Noise ratio of the node labels (Symmetric) |0.2| 0.4|  0.6| 0.8|
>     |-----|----|---|---|---|
>     |Noise ratio of the PI labels| 0.008| 0.021| 0.031|0.036|
>
>     |Noise ratio of the node labels (Asymmetric) |0.2| 0.4|  0.6| 0.8|
>     |-----|----|---|---|---|
>     |Noise ratio of the PI labels| 0.012|0.021 | 0.029|0.035|
>
>
>     For CiteSeer,
>     |Noise ratio of the node labels (Symmetric) |0.2| 0.4|  0.6| 0.8|
>     |-----|----|---|---|---|
>     |Noise ratio of the PI labels|0.009 |0.017 | 0.024|0.034|
>
>     |Noise ratio of the node labels (Asymmetric) |0.2| 0.4|  0.6| 0.8|
>     |-----|----|---|---|---|
>     |Noise ratio of the PI labels|0.009 |0.016 | 0.025|0.031|
>
> - In addition, we strongly agree with your statement that _smaller PI label noise rate does not necessarily imply that it is easier to estimate PI_. In fact, we emphasize that PI estimation is important and challenging in the original submission (**paragraphs 3,4,5 of the Introduction**). Therefore, our proposed PI-GNN makes three specific designs to help PI estimation.
>     - We introduce an end-to-end confidence-aware PI label estimation branch that dynamically estimates PI labels with the help of graph structure.
>     - Moreover, we derive PI labels with the predictive confidence from a PI label estimation network to quantify the reliability of such graph structure.
>     - Additionally, we explore a novel decoupled training approach and propose to decouple the PI label estimation procedure from training with noisy node labels to prevent corruption on the estimated PI labels.
>
>     To verify whether using a larger PI estimation network $f_e$ is helpful, we increase the model size of $f_e$ (from 0.02M to 0.09M, 0.15M, 0.36M) and test on Cora dataset with GCN as the model architecture:
>     |    Noise type    | No Noise|   Symmetric Noise|   | | |Asymmetric Noise   |  | | |
>     | ------ | ----- | ----- |----- | ------ | ----- | ----- |----- | ------ | ----- |
>     |Noise ratio |0.0 | 0.2 | 0.4 | 0.6 | 0.8 |  0.2 | 0.4 | 0.6 | 0.8 |
>     |PI-GNN (0.09M) | 0.782(0.00)|0.728(0.03) | 0.661(0.04)| 0.507(0.05)|0.288(0.05)|**0.734(0.02)**|0.572(0.05)|0.317(0.03)| 0.240(0.02)|
>     |PI-GNN (0.15M) | 0.774(0.01)|0.733(0.05) |  0.652(0.03)|0.510(0.04)| 0.293(0.05)| 0.730(0.03)|0.569(0.01)|0.348(0.03)|0.244(0.03)|
>     |PI-GNN (0.36M) | **0.789(0.01**)| 0.732(0.05) |  **0.671(0.03)**| 0.493(0.06)| 0.281(0.03)| 0.728(0.03) | 0.584(0.04)|0.341(0.03)|**0.249(0.05)**|
>     |PI-GNN (0.02M, Ours) |0.780(0.01) | **0.739(0.02)**| 0.664(0.03)| **0.515(0.03)**| **0.296(0.05)** |0.723(0.03) |**0.587(0.07)** |**0.350(0.07)** |0.232(0.06)|
>
>     where the gap incurred by using larger $f_e$ is not obvious (0.36M vs. 0.02M).

---

> > ### Author Response · Authors · 2023-03-11
> > **Response to Reviewer kfuH-part II**
> >
> > **A2. Subgraph sampling on smaller graphs**
> >
> > We test the performance of PI-GNN (GCN as the backbone) after using subgraph sampling on smaller graphs, i.e., Cora and CiteSeer datasets. Specifically, we sample 15 and 10 neighbors for each node in the 1st and 2nd layer of the GNN and set the batch size to 128. The results are added in **Appendix Section Q** of the updated submission. We also show the new table below:
> >
> > For Cora:
> > |    Noise type    | No Noise|   Symmetric Noise|   | | |Asymmetric Noise   |  | | |
> > | ------ | ----- | ----- |----- | ------ | ----- | ----- |----- | ------ | ----- |
> > |Noise ratio |0.0 | 0.2 | 0.4 | 0.6 | 0.8 |  0.2 | 0.4 | 0.6 | 0.8 |
> > |PI-GNN (Subgraph sampling) | 0.771(0.01)  |0.728(0.03)  |0.651(0.03)  |**0.523(0.05)** | **0.311(0.02)**  |0.708(0.03)  |0.573(0.07)  |0.349(0.01)  |**0.246(0.07)**|
> > |PI-GNN (Ours) |**0.780(0.01)** | **0.739(0.02)**| **0.664(0.03)**| 0.515(0.03)| 0.296(0.05) |**0.723(0.03)** |**0.587(0.07)** |**0.350(0.07)** |0.232(0.06)|
> >
> > For CiteSeer:
> >
> > |    Noise type    | No Noise|   Symmetric Noise|   | | |Asymmetric Noise   |  | | |
> > | ------ | ----- | ----- |----- | ------ | ----- | ----- |----- | ------ | ----- |
> > |Noise ratio |0.0 | 0.2 | 0.4 | 0.6 | 0.8 |  0.2 | 0.4 | 0.6 | 0.8 |
> > |PI-GNN (Subgraph sampling) | 0.674(0.01) |0.639(0.04) |0.579(0.07)| 0.430(0.09) |**0.258(0.05)** |0.618(0.04) |0.528(0.04) |0.348(0.03) |**0.250(0.08)** |
> > |PI-GNN (Ours) |**0.684(0.03)**  |**0.642(0.03)** | **0.591(0.03)**  |**0.432(0.07)** | 0.245(0.05) | **0.628(0.03)**  |**0.531(0.06)** | **0.353(0.06)** | 0.238(0.06)|
> >
> > where we can observe that applying subgraph sampling is less effective than using the entire graph as the input on smaller noise ratios.
> >
> > **A3. Comparison on OGB dataset**
> >
> > As suggested, we have implemented all the baselines on OGB-arxiv dataset and compared with PI-GNN using subgraph sampling. The results are added in **Appendix Section R** of the updated submission. We also show the new table below:
> >
> > |Noise type | Symmetric Noise| |  Asymmetric Noise| |
> > |------ | ----- | ----- |----- |----- |
> > |  Noise ratio |0.4 |0.6 |0.4 |0.6 |
> >   |Decoupling| 0.385(0.09)| 0.347(0.05) | 0.411(0.04) |0.029(0.01)|
> >    |  GCE|0.451(0.05) | 0.407(0.02) | 0.391(0.06) | 0.057(0.03) |
> >    | APL | 0.412(0.05)| 0.375(0.05) | 0.399(0.06) | 0.062(0.01) |
> >    |Co-teaching|0.461(0.04) | 0.403(0.04) | 0.410(0.05) | 0.038(0.01) |
> >    |  LPM-1|0.450(0.01) | 0.397(0.03) | 0.439(0.06) | 0.056(0.01)|
> >    |T-Revision|  0.417(0.04)|  0.409(0.05) | 0.427(0.05) | **0.071(0.04)** |
> >   | DivideMix| 0.448(0.01) | 0.403(0.05) | 0.438(0.05) | 0.041(0.02)|
> >  |PI-GNN (ours)| **0.467(0.03)** |**0.418(0.04)**  |**0.461(0.01)** |0.069(0.01)|
> >
> > where we observe PI-GNN can still outperform all the baselines in most cases.
> >
> >
> > **A4. Manually generated noise**
> >
> > Great point raised! We follow the common practice of the literature in label-noise learning and simulate two noise types in the original submission. We agree that the real-world noisy graphs are natural testbeds for evaluating different robust graph learning algorithms. However, we are not aware of any existing dataset that can be adopted currently for empirical evaluation. We have discussed this limitation in the updated submission (**Conclusion Section**) and will explore on this direction in the future!
> >
> > **A5. Figure glitches**
> >
> > Thanks for capturing this! We have updated the figure 1 of the submission.
> >
> > **A6. Discussions on topics of out-of-distribution node detection and uncertainty prediction**
> >
> > As suggested, we have discussed the mentioned topics with proper citations in **Section 5** of the updated submission.

---

### Review · Reviewer_8ntc · 2023-02-27

**Summary Of Contributions:**

This paper consider graph learning problems (i.e., node classification) where its node labels may be noisy and corrupted. Standard training objective on noisy labels may lead to overfitting and poor generalization. Thus, the author introduced an additional regularized loss function that encouraged two node embeddings to have higher similarity scores if that two nodes are connected and vice versa. In addition to the binary pairwise interactions (PI) label matrix used in the regularization loss, the author also proposed a smoothly estimated PI matrix obtained by via another standalone GNN and shown marginal improvement over the binary PI version.

**Audience:**

Yes

**Broader Impact Concerns:**

No concerns on the ethical implications.

**Claims And Evidence:**

No

**Requested Changes:**

I can not recommend acceptance of this paper, unless the following changes are made:
- (1) Discuss the underlying assumption on graph/label distribution, namely graph homophily, exploited by the PI-GNN. The author need to be aware of the limitation of their PI-GNN framework on certain non-homophily graphs.
- (2) More justification on why using another GNN to estimate the adjacency matrix of graph is a good choice, and compare with other baselines which can also generate smooth estimation of the graph adjacency matrix, such as other matrix decomposition methods.


**Strengths And Weaknesses:**

**Strengths**

- The propose framework, PI-GNN is effective compared to the vanilla GCN and other noisy-robust GNN method.
- The experiment results are comprehensive, covering different 3 GNN architectures, 2 noisy-label schemes with different nosy levels, etc.


**Weaknesses**

- The writing of Section 3 is not clear and some definitions are missing. For example,
(1) what's the definition of $y_{\text{structure}}$? It seems to me its equivalent to the graph adjacency matrix A_{i,j}?
(2) what's the exact regularization loss used in Equation (5)? It seems to be missing in the paper.

- The effectiveness of the proposed method PI-GNN relies on a critical assumption about the underlying graph structure and their label distribution. Namely, the graph need to be *homophilous* (i.e., if two nodes are connected, then with high probability, they should have same node label). However, there are real-world graph datasets that violate the homophily assumption. See more examples in [1,2,3].
In other words, if the graph is non-homophily, then encouraging two connected nodes to have high similarity scores may not help the regularization, but even harm the predictive power of GNN.

- The necessity of smoothly-estimated PI label matrix using another GNN seems questionable. For example, from Table 1 and Figure 3, PI-GNN w/o pc seems to have very similar performance compared to the PI-GNN, where the former is using binary PI label matrix, while the latter learned another GNN to estimate the PI label matrix, and produced an smooth prediction of the binary PI. The author should also consider other conventional methods that generate smooth estimation of the graph adjacency matrix. For example, if we just conduct SVD on the graph adjacency matrix, and obtain the low rank presentation for each node, and use that to generate the smooth estimation, how that compare with the PI-GNN framework that used GNN to do the PI estimation problem?

- Note that learning the GNN $f_e$ for PI estimation in Equation (3) itself is an independent process to $f_t$. Why performing alternative update between $f_e$ and $f_t$ in Algorithm 1? Why not consider first solving $f_e$ up to converge, and use that optimal $f_e^*$ to output the final regularized loss used in Equation (5)?


**Reference**
- [1] Zhu et al. Beyond Homophily in Graph Neural Networks: Current Limitations and Effective Designs. NeurIPS 2020
- [2] Lim et al. Large Scale Learning on Non-Homophilous Graphs: New Benchmarks and Strong Simple Methods. NeurIPS 2021
- [3] Chien et al. Node Feature Extraction by Self-Supervised Multi-scale Neighborhood Prediction. ICLR 2022

---

> ### Author Response · Authors · 2023-03-11
> **Response to Reviewer 8ntc**
>
> We are glad that the reviewer finds our work effective, and with comprehensive ablations. We thank the reviewer for the thorough comments and suggestions, which we address below:
>
>
> **A1. Writing glitches of Section 3**
>
> As suggested, we have revised the writing of Section 3 in the updated submission by removing $y_{\text{structure}}$ and adding a clearer definition of $\mathcal{L}_{\text{reg}}$.
>
> **A2. Assumption on homophilous graphs**
>
> Excellent question here. As suggested, we have updated the submission (**at the end of Section 3**) with the discussion on the underlying assumption of the graph homophily, i.e., the connected nodes are more likely to have the same node labels. We have also included the proper citations of the literature that deals with the non-homophilous graphs. Additional results of the proposed PI-GNN on heterophilous datasets are shown in **Appendix Section E**.
>
>
> **A3. Clarification of using GNN and other alternatives for PI estimation**
>
> Thanks for the suggestions! We would like refer the reviewer to **the paragraph 4 of Section 3.1** in the original submission for the intuitive explanation on the design of the PI estimation. Concretely, the estimated PI label measures the predictive confidence by looking at the closeness between the prediction and the graph structure (i.e., node connectivity). If the predictive confidence becomes far away from the binary node connectivity (0 or 1), then the reliability of such predictions is relatively low, which results in a smoothed PI label and vice versa. We show in **Section 4.5** the ablations of training with different kinds of  PI labels, including _training with the PI labels by comparing the clean node labels (oracle case)_, where using predictive confidence as the PI labels shows competitive performance.
>
> Following the reviewer's suggestion, we estimated the PI labels by performing SVD on the input graph and using the low-rank representations as the estimation results. Suppose the rank of the representation is $r$, we tested the node classification performance under different values of $r$ (i.e., 1, 50, 100, 200, 400, 600, 1000).  Note that we use GCN and Cora as the GNN architecture and the dataset, respectively. During the low-rank approximation, we force the values of the smoothed reconstruction to be greater than 0 and less than 1 by value clipping. The results are updated in **Table 14** of the submission. We also show the new table below:
>
> |    Noise type    |   Symmetric Noise|   | | |Asymmetric Noise   |  | | |
> | ------ |  ----- |----- | ------ | ----- | ----- |----- | ------ | ----- |
> |Noise ratio | 0.2 | 0.4 | 0.6 | 0.8 |  0.2 | 0.4 | 0.6 | 0.8 |
> |GCN |0.722(0.03) |0.613(0.07)| 0.446(0.06)| 0.285(0.07)| 0.703(0.04) |0.514(0.06) |0.291(0.04) |0.161(0.02)|
> |low-rank approximation-1 | 0.693(0.01)|0.587(0.03)|0.418(0.05)|0.272(0.03)|0.688(0.06)|0.483(0.04)|0.269(0.02) |0.110(0.05) |
> |low-rank approximation-50 |0.698(0.04) |0.593(0.03)|0.436(0.03)|0.269(0.01)|0.679(0.04)|0.502(0.06)|0.271(0.08) |0.132(0.08) |
> |low-rank approximation-100 | 0.712(0.09) |0.603(0.05)|0.427(0.06)|0.277(0.05)|0.673(0.01)|0.515(0.06)|0.286(0.04) |0.149(0.02) |
> |low-rank approximation-200 |  0.732(0.04) |0.607(0.04)|0.474(0.04)|0.282(0.03)|0.694(0.05)|0.547(0.06)|0.294(0.01) |0.173(0.08) |
> |low-rank approximation-400 | 0.734(0.01) |0.611(0.09)|0.496(0.05)|**0.300(0.07)**|0.686(0.01)|0.563(0.02)|0.319(0.00) |0.183(0.05) |
> |low-rank approximation-600 | 0.726(0.02) |0.626(0.02)|0.476(0.03)|0.298(0.03)|0.695(0.04)|0.534(0.07)|0.321(0.03) |0.201(0.03) |
> |low-rank approximation-1000 |0.704(0.05) |0.610(0.08)|0.455(0.04)|0.256(0.07)|0.668(0.02)|0.508(0.05)|0.346(0.07) |0.188(0.09) |
> |PI-GNN (Ours)|**0.739(0.02)**| **0.664(0.03)**| **0.515(0.03)**| 0.296(0.05) |**0.723(0.03)** |**0.587(0.07)** |**0.350(0.07)** |**0.232(0.06)**|
>
>
> where $r=400$ roughly achieves the best performance but still cannot outperform our PI-GNN. The reason might be that low-rank approximation is mainly designed for matrix compression, denoising and completion (https://web.stanford.edu/class/cs168/l/l9.pdf), which cannot capture the uncertainty of the PI labels in essence as in PI-GNN. Moreover, we observe that during low-rank approximation, the values of each node pair will quickly go from negative values to values larger than 1, which is not suitable to be the PI labels for training. Although we directly adopt the clipping approach to force the values to be in the range of 0 and 1, additional curated designs might be more beneficial. Finally, the rank $r$ is an important hyperparameter to tune in the low-rank approximation approaches, which requires extra manual tuning compared to our PI-GNN. We have added the above discussion in the updated paper.

---

> > ### Author Response · Authors · 2023-03-11
> > **Response to Reviewer 8ntc (part II)**
> >
> > **A4. Clarification on the algorithm block**
> >
> > As suggested, we have tried to firstly pretrain the PI label estimation network $f_e$ for 400 epochs. Then we directly apply the pretrained $f_e$ to output the estimated PI labels for training $f_t$. The results on Cora and CiteSeer are shown below:
> >
> > For Cora:
> > |    Noise type    | No Noise|   Symmetric Noise|   | | |Asymmetric Noise   |  | | |
> > | ------ | ----- | ----- |----- | ------ | ----- | ----- |----- | ------ | ----- |
> > |Noise ratio |0.0 | 0.2 | 0.4 | 0.6 | 0.8 |  0.2 | 0.4 | 0.6 | 0.8 |
> > |PI-GNN (Pretrain) | 0.772(0.01)| 0.733(0.03)| **0.672(0.04)**| 0.508(0.06)|**0.319(0.04)**| **0.728(0.04)**|0.569(0.04) | 0.339(0.06) | **0.245(0.06)**|
> > |PI-GNN (Ours) |**0.780(0.01)** | **0.739(0.02)**| 0.664(0.03)| **0.515(0.03)**| 0.296(0.05) |0.723(0.03) |**0.587(0.07)** |**0.350(0.07)** |0.232(0.06)|
> >
> > For CiteSeer:
> >
> > |    Noise type    | No Noise|   Symmetric Noise|   | | |Asymmetric Noise   |  | | |
> > | ------ | ----- | ----- |----- | ------ | ----- | ----- |----- | ------ | ----- |
> > |Noise ratio |0.0 | 0.2 | 0.4 | 0.6 | 0.8 |  0.2 | 0.4 | 0.6 | 0.8 |
> > |PI-GNN (Pretrain) | **0.693(0.01)**|0.631(0.03)| **0.606(0.04)**|**0.452(0.04)** | 0.229(0.05)|0.604(0.04) | 0.511(0.04)| 0.338(0.02)| **0.242(0.06)** |
> > |PI-GNN (Ours) |0.684(0.03)  |**0.642(0.03)** | 0.591(0.03)  |0.432(0.07) | **0.245(0.05)** | **0.628(0.03)**  |**0.531(0.06)** | **0.353(0.06)** | 0.238(0.06)|
> >
> > where pretraining the PI label estimation network and freezing it during training the node classification model achieves a similar classification performance compared to our training scheme in the original submission. We have added the discussion in the **Appendix Section N** of the updated submission.

---

### Review · Reviewer_AoRd · 2023-03-04

**Summary Of Contributions:**

The paper proposes a novel method for teaching Graph Neural Networks (GNNs) to accurately classify nodes under severely noisy labels. The authors propose a pairwise training method that leverages structural pairwise interactions (PI) between nodes to improve label noise learning.

The paper is well-structured and provides a clear introduction to the problem of noisy labels in graph learning applications. The authors highlight the importance of this problem and note that it is currently underexplored. They then introduce their proposed method, which builds on previous work in supervised metric learning and unsupervised contrastive learning.

The experimental results presented in the paper are comprehensive and provide strong evidence for the effectiveness of the proposed method. The authors compare their method with several baselines on test datasets Cora and CiteSeer with symmetric and asymmetric noise (noise rate ε = 0.4, 0.6). From Table 2, it is evident that PI-GNN outperforms different baselines with a considerable margin, improving the classification accuracy by 5.2% on Cora under symmetric noise (ε = 0.6) compared to the best baseline.

Moreover, PI-GNN is able to outperform LPM-1, which relieves the strong assumption that auxiliary clean node labels are available. The authors also compare their method with traditional graph semi-supervised learning approaches and on more datasets in Appendix Section D and Section J, respectively.

Overall, this paper presents a novel approach to address an important problem in graph learning applications - noisy labels. The proposed pairwise training method leverages structural pairwise interactions between nodes to improve label noise learning and provides strong experimental evidence for its effectiveness compared to several baselines.


**Audience:**

Yes

**Claims And Evidence:**

Yes

**Requested Changes:**

Cons:
- The proposed method may require more computational resources compared to traditional graph semi-supervised learning approaches due to its reliance on pairwise interactions between nodes. Thus, please provide complexity analysis and experiments.

- The paper does not provide a detailed analysis of how sensitive the proposed method is to different hyperparameters or initialization methods. Further research may be needed to explore these aspects.

- While the experimental results presented in the paper are promising, it is unclear how well PI-GNN would perform on larger or more complex datasets. Further research may be needed to explore this aspect as well.

- Some most relevant papers are not cited and compared. Such as:
Learning to Drop: Robust Graph Neural Network via Topological Denoising.
The 14th ACM International WSDM Conference (WSDM'21), 2021

**Strengths And Weaknesses:**

Pros:
- The proposed pairwise training method leverages structural pairwise interactions (PI) between nodes to improve label noise learning, which is a novel approach to address an important problem in graph learning applications.
- The experimental results presented in the paper demonstrate the effectiveness of PI-GNN compared to several baselines on different datasets and GNN architectures.
- The proposed confidence-aware PI estimation model adaptively estimates the PI labels, which are defined as whether the two nodes share the same node labels. This approach can help improve the accuracy of noisy node classification on graphs.
- The decoupled training approach leverages the estimated PI labels to regularize a node classification model for robust node classification. This approach can help improve the robustness of GNNs to noisy labels.

---

> ### Author Response · Authors · 2023-03-11
> **Response to Reviewer AoRd**
>
> We are really encouraged that you recognize our method to be novel, well-structured, and with effective and extensive empirical results.
>
> Your summary and comments are insightful and spot-on :)
>
> **A1. Complexity analysis and experiments**
>
> Excellent question here. We want to refer the reviewer to **the end of Section 3** in the submission where we provide the time complexity analysis for the PI label estimation model $f_e$. Specifically, assume $d$ is the dimension of the node embedding, $|E|,|V|$ are the number of edges and nodes of the graph and $T,L$ are the number of iterations and the layers, the time complexity is $\mathcal{O}\left(T L|E| d+T L |V| d^{2} + T |V|^2d^2\right)$. For big graph datasets with large $|E|$ and $|V|$, we perform subgraph sampling [1] in the implementation to reduce the time complexity. Moreover, we also provide the empirical evaluation on the running time for our PI-GNN in **Appendix Section I**, where our approach does not incur more inference cost than a vanilla GNN model.
>
> [1] William L. Hamilton, Rex Ying, Jure Leskovec, Inductive Representation Learning on Large Graphs, NIPS 2017.
>
>
> **A2. Sensitivity analysis to different hyperparameters and initializations**
>
> - For analysis on the hyperparameters, we would like to refer the reviewer to **Figure 5** of the submission, which shows the sensitivity ablations on two key hyperparameters of PI-GNN, i.e., the pretraining epochs for the PI label estimation network $K$ and the weight $\beta$ for the regularization loss. All the other training configurations, such as the learning rate, weight decay, model architecture, etc., are kept the same as the default values in the torch-geometric library (https://github.com/pyg-team/pytorch_geometric/blob/master/examples).
>
> - For the analysis on the model initialization, we add a new table (**Table 12**) in the updated paper, where we tested 4 additional initilizations, i.e., "uniform initialization", "normal initialization", "constant initialization", and "kaiming initialization". We compare the performance of PI-GNN using the "glorot initialization" (which is the initialization we used in the submission). As in the original submission, we use Cora and Citeseer datasets and set the noise ratio to 0.6 (both symmetric and non-symmetric noise). The results demonstrate that the performance is not sensitive to different model initializations, where the biggest gap among all the model initializations is smaller than 3%. We also show the new table below:
>
>     |    Noise type    | Symmetric Noise|     Asymmetric Noise   |
>     | ------ | ----- | ----- |
>     |Noise ratio |0.6 |0.6|
>     | | Test dataset: Cora / CiteSeer| Test dataset: Cora / CiteSeer|
>      |Uniform Initialization| 0.514(0.05) / 0.443(0.05) | 0.364(0.02) / 0.321(0.04) |
>      | Normal Initialization| 0.548(0.03) / 0.426(0.05) | 0.361(0.05) / 0.345(0.01)|
>      | Constant Initialization | 0.504(0.01) / 0.413(0.04) | 0.339(0.03) / 0.341(0.04) |
>      | Kaiming Initialization| 0.503(0.04) / 0.429(0.03) | 0.333(0.02) / 0.372(0.09)|
>     |**Glorot Initialization (Ours)** | 0.515(0.03) / 0.432(0.07) | 0.350(0.07) / 0.353(0.06) |
>
>
> **A3. Results on larger and more complex datasets**
>
> Another great question! We have tested our proposed approach on the large OGB-arxiv dataset (c.f. Table 1 in the submission). During rebuttal, we have tested PI-GNN on an even larger OGB-products dataset, which has 2,449,029 nodes with 61,859,140 edges and thus is much larger than the OGB-arxiv dataset (169,343 nodes and 1,166,243 edges). The results are added in the updated submission (c.f. **Table 13** in the Appendix). We also show the new table below:
>
> |    Noise type    | No Noise|   Symmetric Noise|   | | |Asymmetric Noise   |  | | |
> | ------ | ----- | ----- |----- | ------ | ----- | ----- |----- | ------ | ----- |
> |Noise ratio |0.0 | 0.2 | 0.4 | 0.6 | 0.8 |  0.2 | 0.4 | 0.6 | 0.8 |
> |GCN |**0.732(0.04)** |0.709(0.01) |0.661(0.09) |0.612(0.07) |0.471(0.06) |0.689(0.03) |0.635(0.03)| 0.102(0.03) |0.053(0.01)|
> |PI-GNN w/o pc |0.711(0.03)| 0.721(0.04)| 0.669(0.03) |0.631(0.06) |0.487(0.09)| 0.700(0.09)| 0.641(0.04) |0.136(0.01)| 0.110(0.04)|
> |PI-GNN |0.727(0.05) |**0.738(0.06)**| **0.677(0.06)** |**0.658(0.03)** |**0.506(0.05)**| **0.719(0.06)** |**0.667(0.03)** |**0.196(0.06)** |**0.153(0.04)**|
>
>
> where our PI-GNN can still demonstrate promise compared to the vanilla GCN model and PI-GNN w/o predictive confidence.
>
> **A4. Missing related works**
>
> Thank you for the suggestions. The suggested paper proposed PTDNet, a parameterized topological denoising network, to improve the robustness and generalization performance of GNNs by learning to _drop task-irrelevant edges_. The main difference is that PI-GNN deals with the situation where the node labels are corrupted while the mentioned paper deals with the noisy edges. We have updaed the submission with proper discussion and citation (c.f. **Section 5**).

---

> > ### Comment · Reviewer_AoRd · 2023-04-16
> > **Concerns addressed**
> >
> > My concerns have been fully addressed. Thanks for the response.

---

### Author Response · Authors · 2023-03-11
**General response**

We thank all the reviewers for their time and valuable comments. We have addressed the reviewers’ comments and concerns in individual responses to each reviewer. The reviews allowed us to strengthen our manuscript and the major changes made (highlighted in blue) are summarized below:

+ [AoRd] Added sensitivity analysis to model initialization.
+ [AoRd] Added results on a larger graph dataset.
+ [AoRd, kfuH] Added discussion on related works.
+ [8ntc, kfuH] Fixed writing glitches.
+ [8ntc] Clarified the assumption on homophilous graphs.
+ [8ntc] Added experiments on PI estimation using low-rank approximation.
+ [8ntc] Added comparison among different training schemes.
+ [kfuH] Clarified the motivation for using PI estimation.
+ [kfuH] Added results on using subgraph sampling on smaller graphs.
+ [kfuH] Added comparison with baselines on the OGB dataset.
+ [kfuH] Clarified the manually generated noise.

---

### Decision · Action_Editors · 2023-04-28

**Recommendation:** Accept as is

**Comment:**

The paper proposes a graph learning method for the case where node labels can be noisy, and shows its good performance.  The reviewers' requests, including improving clarity, additional experiments, and discussion on missing related work, have been well addressed.

In the revision, there are wrong uses of citet and citep for example in the end of Section 3.  Please correct them along with typos for the final version.

**Audience:**

Yes, the problem tackled is an important problem for graph learning.

**Claims And Evidence:**

Yes, the claims are supported by experiments.